# Instance-Optimal Pure Exploration for Linear Bandits on Continuous Arms

**Sho Takemori** [1]  **Yuhei Umeda** [1]  **Aditya Gopalan** [2]

## Abstract

This paper studies a pure exploration problem with linear bandit feedback on continuous arm sets, aiming to identify an $\epsilon$-optimal arm with high probability. Previous approaches for continuous arm sets have employed instance-independent methods due to technical challenges such as the infinite dimensionality of the space of probability measures and the non-smoothness of the objective function. This paper proposes a novel, tractable algorithm that addresses these challenges by leveraging a reparametrization of the sampling distribution and projected subgradient descent. However, this approach introduces new challenges related to the projection and reconstruction of the distribution from the reparametrization. We address these by focusing on the connection to the approximate Carathéodory problem. Compared to the original optimization problem on the infinite-dimensional space, our method is tractable, requiring only the solution of quadratic and fractional quadratic problems on the arm set. We establish an instance-dependent optimality for our method, and empirical results on synthetic environments demonstrate its superiority over existing instance-independent baselines.

## 1. Introduction

The stochastic linear bandit problem (Dani et al., 2008; Jedra & Proutiere, 2020) is a well-studied online decision-making problem that has been widely applied to real-world problems (Lattimore & Szepesvári, 2020). While most research focuses on minimizing cumulative regret, this paper considers a pure exploration problem, specifically the $\epsilon$-BAI (best arm identification) problem. Unlike regret-minimizing algorithms, optimal pure exploration algorithms achieve an exponentially decaying probability of misidentifying

[1]Fujitsu Limited, Kawasaki, Japan [2]Indian Institute of Science, Bengaluru 560012, India. Correspondence to: Sho Takemori <takemori.sho@fujitsu.com>.

*Proceedings of the 42nd International Conference on Machine Learning*, Vancouver, Canada. PMLR 267, 2025. Copyright 2025 by the author(s).

an $\epsilon$-optimal arm (cf. (Bubeck et al., 2011, Theorem 1), Lemma 4.1, (Banerjee et al., 2023, Theorem 2.2)), where $\epsilon > 0$ is a given error tolerance. Furthermore, this paper considers a *continuous arm set* $\mathcal{X} \subset \mathbb{R}^d$ (a compact subset of $\mathbb{R}^d$) and aims to optimize sampling distributions to minimize the posterior probability of misidentification under a *Bayesian* reward model. Despite extensive existing research on BAI problems, as we detail below, the $\epsilon$-BAI problem on continuous arm sets is an open problem.

The pure exploration problem with linear bandit feedback in the *finite-armed setting* has been extensively studied, with instance-optimal algorithms proposed for the $(\epsilon, \delta)$-PAC learning setting (Jourdan & Degenne, 2022), aiming to identify an $\epsilon$-optimal arm with probability at least $1 - \delta$. These algorithms typically solve an optimization problem over the probability simplex $\Delta_{K-1} \subset \mathbb{R}^K$, where $K$ is the number of arms; for example, Lazy Track-and-Stop (Jedra & Proutiere, 2020) employs the Frank-Wolfe algorithm. Naively applying these finite-armed algorithms to the continuous arm setting via discretization of the arm set $\mathcal{X}$ is computationally intractable, as the number of discretization points $K$ can grow exponentially with the dimension $d$.

Although most existing works for pure exploration problems focused on the finite-armed setting, there are few notable exceptions. Jedra & Proutiere (2020) studied a linear pure exploration problem with the fixed confidence setting and proposed an (order-wise) optimal algorithm when the arm set $\mathcal{X}$ is the unit sphere (along with the case of the finite-armed setting). However, their analysis shows that a uniform exploration is an (order-wise) optimal sampling rule and such an exploration would not be optimal for general continuous arm sets. Bhat & Amballa (2022) also studied the $(\epsilon, \delta)$-PAC learning problem with the linear bandit setting for general continuous arm sets. However, their method is also a uniform exploration and theoretical result (Bhat & Amballa, 2022, Theorem 4.4, Eq. (10)) shows an upper bound of the sample complexity involves the dimension $d$ and $\epsilon$ rather than a problem instance, i.e., their analysis is *instance-independent*. Therefore, even in the case of linear bandits, to the best of our knowledge, it is an open problem to develop a trackable algorithm for a pure exploration problem with linear bandit feedback with an *instance-dependent* optimality.

As we detail in Section 4, developing a tractable $\epsilon$-BAI algorithm is challenging because finding an optimal sampling strategy requires optimization over the space $\mathcal{P}(\mathcal{X})$ of probability measures on $\mathcal{X}$. This optimization problem (defined as Eq. (4)) is highly non-trivial to solve due to (i) the *infinite-dimensionality* of $\mathcal{P}(\mathcal{X})$ when the arm set $\mathcal{X}$ is continuous, and (ii) the *non-smoothness* of the objective function (4) due to the inner supremum. Furthermore, bandit feedback obscures the true objective function. This paper addresses these challenges by reparameterizing $\pi \mapsto V(\pi)$ for $\pi \in \mathcal{P}(\mathcal{X})$, where $V(\pi) = \mathbb{E}_{x \sim \pi} \left[ xx^\top \right] \in \mathbb{R}^{d \times d}$ is the covariance matrix, and adapting a projected subgradient descent (PSGD) method (cf. (Bubeck, 2015, Chapter 3.1)). This approach updates a symmetric matrix $V_t \in \mathbb{R}^{d \times d}$ such that $V_t = V(\pi_t)$, where $\pi_t$ is an arm's sampling distribution at round $t$. However, it introduces new challenges: (iii) projecting onto the space $\mathcal{V}(\mathcal{X}) := \{V(\pi) : \pi \in \mathcal{P}(\mathcal{X})\}$, and (iv) reconstructing a distribution $\pi_t$ from a given matrix $V_t$. We address these by leveraging a connection to the approximate Carathéodory problem (Combettes & Pokutta, 2023), which seeks a convex combination approximating a given point in a convex set. Analogously to the approximate Carathéodory problem, we resolve (iii) and (iv) using an adapted Frank-Wolfe method (Algorithm 2).

To summarize our contributions, we briefly describe the problem setting (a more formal description is given in Section 3). We assume a Bayesian linear reward model, i.e., for each $t = 1, 2, \ldots$, for a selected arm $x_t \in \mathcal{X}$, a learner observes a noisy reward $y_t = f(x) + \omega_t$, where $\omega_t$ is a zero-mean noise, $f(x)$ is an unknown random function given as $f(x) = \theta^f \cdot x$ and $\theta^f$ is an unknown random vector sampled from a prior distribution $\mathcal{N}(\theta_0, \Sigma_0)$ with $\theta_0 \in \mathbb{R}^d, \Sigma_0 \in \mathbb{R}^{d \times d}$ when the round starts. The learner samples an arm $x_t$ using a distribution $\pi_t \in \mathcal{P}(\mathcal{X})$, where $\pi_t$ is a distribution computed using observations up to round $t - 1$, and we call $(\pi_t)_{t \geq 1}$ a *sampling rule*. For each round $t \geq 1$, after observing $y_t$, a learner recommends an arm $\zeta_t \in \mathcal{X}$ that estimates an $\epsilon$-optimal arm. The objective of the learner to minimize the posterior probability $P_t(\zeta_t \notin \mathcal{X}^*(\epsilon))$ of misidentification, where $\mathcal{X}^*(\epsilon)$ is the set of $\epsilon$-optimal arms given as $\{x \in \mathcal{X} : f(x) > \sup_{\xi \in \mathcal{X}} f(x) - \epsilon\}$ and $P_t$ denotes the posterior probability given observations up to round $t$. For conciseness, we provide informal theoretical statements of our theoretical results and refer to Lemma 4.1 and Theorem 6.2 for more formal statements (All the omitted proofs can be found in Appendix F).

1. **Formalization of the objective.** To formalize our optimization objective regarding sampling rules, we show that $\lim_{t \to \infty} \frac{1}{t} \log P_t(\zeta_t \notin \mathcal{X}^*(\epsilon))$ can be written as a function of $\overline{\pi}_\infty = \lim_{t \to \infty} \overline{\pi}_t$ and $\zeta_\infty = \lim_{t \to \infty} \zeta_t$, where $\overline{\pi}_t := \frac{1}{t} \sum_{s=1}^{t} \pi_s$ (Lemma 4.1).

2. **A trackable algorithm.** We propose a novel, tractable

algorithm termed PMCA in Section 5.1; a method for Posterior error Minimization for a general Continuous Arm set based on the PSGD (projected subgradient descent method) that has an interesting connection to the Approximate Carathéodory problem. More precisely, our algorithm assumes computation oracles for *quadratic and fractional quadratic* objectives on the arm set $\mathcal{X}$, which can be implemented by standard optimization library unlike the original optimization problem on $\mathcal{P}(\mathcal{X})$.

3. **An instance-dependent (or asymptotic) optimality of the sampling strategy.** We show that our proposed method is asymptotically optimal (Theorem 6.2), i.e., for any given recommendation rule $(\zeta_t)_{t \geq 1}$, the sampling rule $(\pi_t)_{t \geq 1}$ of PMCA minimizes the posterior probability asymptotically, i.e., it minimizes $\lim_{t \to \infty} \frac{1}{t} \log P_t(\zeta_t \notin \mathcal{X}^*(\epsilon))$.

4. **Experiments.** To demonstrate the effectiveness of our method, in simple, synthetic environments, we empirically compare our method with baselines (including MVR (Vakili et al., 2021), which is instance-independent optimal) and the empirical results demonstrate its superiority over existing instance-independent baselines (Section 7).

As we shall clarify in Section 6, we provide our main theoretical result (Theorem 6.2) for a *given* recommendation rule $(\zeta_t)_{t \geq 1}$ satisfying Eq. (7). Our analysis implies that a pair $((\pi_t)_{t \geq 1}, (\zeta_t)_{t \geq 1})$ is (asymptotically) instance-optimal *if* the recommendation rule $(\zeta_t)_{t \geq 1}$ is asymptotically optimal (we refer to Section 4.3 for more details for choices of a recommendation rule). In this work, assuming a recommendation rule $(\zeta_t)_{t \geq 1}$ is given, we focus on the optimization of a sampling rule $(\pi_t)_{t \geq 1}$, which we believe is the most challenging aspect in the continuous arm setting. Nevertheless, we emphasize that our analysis is instance-dependent unlike existing pure exploration methods for the continuous arm setting (such as MVR Vakili et al. (2021)).

## 2. Related Work

This paper studies the $\epsilon$-BAI problem with linear bandit feedback on a *continuous* arm set $\mathcal{X} \subset \mathbb{R}^d$ and our objective is to minimize the posterior probability $P_t(\zeta_t \notin \mathcal{X}^*(\epsilon))$ of misidentification under a Bayesian linear reward model. As we mentioned in the introduction, Bhat & Amballa (2022) studied a $(\epsilon, \delta)$-PAC learning problem with linear bandit feedback on a continuous arm set, where the objective is to minimize the stopping time of an algorithm that returns an $\epsilon$-optimal arm with confidence $1 - \delta$. The $(\epsilon, \delta)$-PAC learning problem with linear bandit feedback has been well-studied especially for the finite-armed setting (Jourdan & Degenne, 2022; Jedra & Proutiere, 2020; Soare et al., 2014;

Xu et al., 2018; Fiez et al., 2019; Degenne et al., 2019; Mohammadi Zaki & Gopalan, 2022). Specifically, Jourdan & Degenne (2022) studied the transductive linear bandits (a more general setting than linear bandits), proposed an $(\epsilon, \delta)$-PAC learning algorithm, and discussed asymptotic optimality as $\delta \to 0$ (i.e., a limit to the case of 100% confidence). Although our objective is to minimize $P_t(\zeta_t \notin \mathcal{X}^*(\epsilon))$, our theoretical analysis is parallel to that of the $(\epsilon, \delta)$-PAC learning setting, i.e., we discuss asymptotic optimality of our algorithm as $t \to \infty$ (Lemma 4.1, Theorem 6.2) and discuss a stopping rule that ensures the posterior probability of misidentification is less than $\delta$ (Section 3, Lemma 4.2). We also note that an asymptotic analysis of the posterior probability of misidentification is standard in the finite-armed setting (Li et al., 2024; Russo, 2016).

Despite extensive existing research for pure exploration problems with linear bandit feedback, most papers focus on the case of finite-armed settings due to the difficulties mentioned in the introduction. To the best of our knowledge, it is an open problem to develop a trackable algorithm for a pure exploration problem on a general continuous arm set with an *instance-dependent* optimality. Other than linear bandit literature, Vakili et al. (2021) proposed an optimal algorithm pure exploration (simple regret minimization) called MVR (maximum variance reduction) for Gaussian process bandits (or Bayesian optimization). However, their analysis is also an instance-independent analysis, and MVR is non-adaptive and selects an arm that minimizes the posterior variance $\sigma_t$. Shekhar & Javidi (2022) provided instance-dependent analysis of kernelized bandits, however, they focused on cumulative regret minimization. We discuss more related works in Appendix C.

## 3. Problem Formulation

**Bayesian Linear Reward Model.** We assume that $\mathcal{X}$ is a compact subset of $\mathbb{R}^d$ and consider a Bayesian linear reward function $f : \mathcal{X} \to \mathbb{R}$. We suppose $\overline{k} > 0$ satisfies $\sup_{x \in \mathcal{X}} \|x\|^2 \leq \overline{k}$. More formally, the random linear function $f$ is defined as $f(x) = \theta^f \cdot x$, where $a \cdot b$ is the standard Euclidean inner product, and $\theta^f$ is a random vector that follows a prior distribution $\theta^f \sim \mathcal{N}(\theta_0, \Sigma_0)$, where $\theta_0 \in \mathbb{R}^d$ and $\Sigma_0 \in \mathbb{R}^{d \times d}$ is a positive definite symmetric matrix. By applying an affine transformation of $\mathcal{X}$, without loss of generality, we may assume $\theta_0 = 0_d$ and $\Sigma_0 = 1_d$, where $0_d$ and $1_d$ are the zero vector and identity matrix of size $d$, respectively. In the rest of the paper, we assume that the prior distribution of $\theta^f$ is given to be $\mathcal{N}(0_d, 1_d)$.

**Sampling Rule.** For each time step $t = 1, 2, \ldots$, a learner selects an arm $x_t \in \mathcal{X}$, where $x_t$ is sampled from a probability distribution $\pi_t = \pi_t(\cdot | x_1, y_1, x_2, y_2, \ldots, x_{t-1}, y_{t-1}) \in \mathcal{P}(\mathcal{X})$. Then, after committing an action $x_t$ the learner observes a random reward $y_t = f(x_t) + \omega_t$ and where for

a positive constant $\lambda > 0$, $\{\omega_t\}_t$ is an i.i.d sequence of noise random variable with $\omega_t \sim \mathcal{N}(0, \lambda)$, which is independent of $f$. We denote by $\mathcal{F}_t$ the $\sigma$-algebra generated by $x_1, y_1, \ldots, x_t, y_t$. The posterior reward function $f_t$ conditioned on $\mathcal{F}_t$ is given as $f_t(x) = \theta_t^f \cdot x$, where $\theta_t^f$ follows a normal distribution $\theta_t^f \sim \mathcal{N}(\mu_t, \Sigma_t^{-1})$, and $\Sigma_t$ and $\mu_t$ are given as $\Sigma_t = \lambda 1_d + \sum_{s=1}^t x_s x_s^\top$, $\mu_t := \Sigma_t^{-1} \sum_{s=1}^t y_s x_s$ (this is a special case of (Kanagawa et al., 2018, Theorem 3.1)). We denote by $P_t$ the posterior probability measure conditioned on $\mathcal{F}_t$.

**Recommendation Rule and Objective of the Learner.** In each round $t$, after observing $y_t$, the learner recommends an arm $\zeta_t \in \mathcal{X}$ that estimates $\zeta_t$ an $\epsilon$-optimal arm at time step $t$. More formally, $\zeta_t$ is a $\mathcal{F}_t$-measurable random variable. Here, the set of $\epsilon$-optimal arms $\mathcal{X}^*(\epsilon)$ is defined as $\{x \in \mathcal{X} : f(x) > \sup_{\xi \in \mathcal{X}} f(\xi) - \epsilon\}$. For example, the greedy recommendation rule is defined by $\zeta_t = \mathrm{argmax}_{x \in \mathcal{X}} \mu_t(x)$, where $\mu_t$ is the posterior mean of $f$. This recommendation rule is used for an existing method for simple regret minimization (Vakili et al., 2021).

The objective of the learner is to minimize the posterior probability of misidentification $P_t(\zeta_t \notin \mathcal{X}^*(\epsilon))$, which depends on the sampling rule $(\pi_t)_{t \geq 1}$ and recommendation rule $(\zeta_t)_{t \geq 1}$. In this paper, for any given recommendation rule $(\zeta_t)_{t \geq 1}$, we aim to seek an optimal sampling rule $(\pi_t)_{t \geq 1}$. As we discuss in Section 4, the optimality involves an optimization problem on the space of probability measures $\mathcal{P}(\mathcal{X})$, which is a highly non-trivial problem due to the infinite dimensionality of $\mathcal{P}(\mathcal{X})$ and non-smoothness of the objective (4). Moreover, we discuss choices of recommendation rule $(\zeta_t)_{t \geq 1}$ in Section 4.3.

**Stopping Rule.** In the $(\epsilon, \delta)$-PAC learning problem, an algorithm stops if it ensures the error probability is less than a given threshold $\delta$. In our problem setting (i.e., the $\epsilon$-BAI problem under a Bayesian reward model), assuming observations up to round $t$, the posterior probability $P_t(\zeta_t \notin \mathcal{X}^*(\epsilon))$ of misidentification is known to the learner, i.e., more formally it is a $\mathcal{F}_t$-measurable random variable. Therefore, instead of conducting the GLLR (generalized log-likelihood ratio) test (Jedra & Proutiere, 2020), it is sufficient to compute $P_t(\zeta_t \notin \mathcal{X}^*(\epsilon))$. However, an actual computation of $P_t(\zeta_t \notin \mathcal{X}^*(\epsilon))$ is non-trivial due to the continuous arm set setting. Hence, we provide a computable upper bound of $P_t(\zeta_t \notin \mathcal{X}^*(\epsilon))$ that asymptotically matches $P_t(\zeta_t \notin \mathcal{X}^*(\epsilon))$ (Lemma 4.2). More precisely, for any explicitly computable upper bound $u_t$ of the conditional probability of misidentification and $\delta \in (0, 1)$, we refer to the stopping rule by the stopping time defined as the minimum $t$ satisfying $u_t \leq \delta$. In the experiment section (Section 7), we compare algorithms in terms of this computable upper bound.

**Assumptions and Notations.** For a symmetric matrix $V \in$

$\mathbb{R}^{d \times d}$, we denote $V \succ 0$ if $V$ is positive definite. For a symmetric matrix $V$, we denote by $\lambda_{\min}(V)$ the minimum eigenvalue of $V$, and define $\|x\|_V$ as $\sqrt{x^\top V x}$ for $x \in \mathbb{R}^d$ if $V \succ 0$. For a matrix $V$, $\|V\|_F$ denotes the Frobenius norm. We denote by $\mathcal{P}(\mathcal{X})$ the space of Borel probability measures on $\mathcal{X}$, which is a general class of probability measures. For $\pi \in \mathcal{P}(\mathcal{X})$, we define $V(\pi) = \mathbb{E}_{x \sim \pi}\left[xx^\top\right]$. The space $\mathcal{V}(\mathcal{X})$ is defined as $\{V(\pi) : \pi \in \mathcal{P}(\mathcal{X})\}$. For $a \in \mathcal{X}$, $\delta(a)$ is the Dirac delta distribution putting all mass on $\{a\}$. We refer to Appendix A for a notation table.

## 4. Posterior Probability of Misidentification

### 4.1. Asymptotic Bounds of the Posterior Probability

For a fixed recommendation rule $(\zeta_t)_{t \geq 1}$, this section introduces results on (asymptotic) lower and upper bounds of the posterior probability of misidentification in terms of the sampling rule $(\pi_t)_{t \geq 1}$ in Lemma 4.1. To do so, we introduce following notations. Here, for a positive-definite matrix $V \in \mathbb{R}^{d \times d}$, $\zeta, \xi \in \mathcal{X}$, and a function $\mu : \mathcal{X} \to \mathbb{R}$, we define

$$\Gamma(V, \xi; \zeta, \mu) := \frac{\|\zeta - \xi\|_{V^{-1}}^2}{(\epsilon + \mu(\zeta) - \mu(\xi))^2}.$$

Then, for a probability distribution $\pi \in \mathcal{P}(\mathcal{X})$, $\zeta \in \mathcal{X}$, $\mu : \mathcal{X} \to \mathbb{R}$, we define

$$\Gamma^*(V(\pi); \zeta, \mu) := \sup_{\xi \in \mathcal{X}} \Gamma(V(\pi), \xi; \zeta, \mu). \quad (1)$$

The following lemma provides a lower bound of the posterior probability of misidentification in terms of the sampling rule $(\pi_t)_{t \geq 1}$ (and recommendation rule $(\zeta_t)_{t \geq 1}$).

**Lemma 4.1** (Asymptotic posterior probability of misidentification). *Let $(\pi_t)_{t \geq 1}$ be a sampling rule and $(\zeta_t)_{t \geq 1}$ be a recommendation rule with $\lim_{t \to \infty} \zeta_t := \zeta_\infty \in \mathcal{X}^*(\epsilon)$, a.s., that is, $\zeta_\infty$ is the a.s. limit of the random variables $(\zeta_t)_{t \geq 1}$. Suppose $\lim_{t \to \infty} \mu_t(x) = f(x)$ for any $x \in \mathcal{X}$ a.s., and $\inf_{t \geq 1} \lambda_{\min}(V(\overline{\pi}_t)) > 0$, where $\overline{\pi}_t := \frac{1}{t} \sum_{s=1}^t \pi_s$. Then, we have the asymptotic bound:*

$$\liminf_{t \to \infty} \frac{1}{t} \log P_t\left(\zeta_t \notin \mathcal{X}^*(\epsilon)\right)$$
$$\geq -\frac{1}{2} \limsup_{t \to \infty} \left(\Gamma^*(V(\overline{\pi}_t); \zeta_\infty, f)\right)^{-1}. \quad (2)$$

*Moreover, we have:*

$$\limsup_{t \to \infty} \frac{1}{t} \log P_t\left(\zeta_t \notin \mathcal{X}^*(\epsilon)\right)$$
$$\leq -\frac{1}{2} \liminf_{t \to \infty} \left(\Gamma^*(V(\overline{\pi}_t); \zeta_\infty, f)\right)^{-1}. \quad (3)$$

Intuitively, this lemma implies that the posterior probability $P_t\left(\zeta_t \notin \mathcal{X}^*(\epsilon)\right)$ exponentially decays as $t$ increases, and

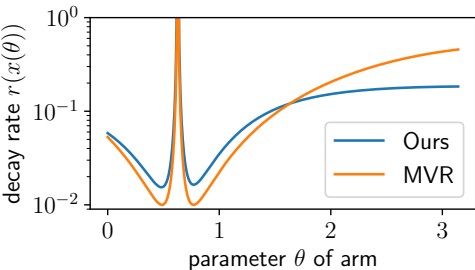

*Figure 1.* A numerical example of $(\Gamma(V(\pi), \xi; \zeta, f))^{-1}$ for $\mathcal{X} \subset \mathbb{R}^2$ and $f(x) := \mu^* \cdot x$ with a fixed reward vector $\mu^* \in \mathbb{R}^2$. An arm in $\mathcal{X}$ is parametrized by $x(\theta) = (\cos(\theta), \sin(\theta))$ with $0 \leq \theta \leq \pi$. The $x$-axis represents the parameter $\theta$ and the $y$-axis represents $r(x(\theta)) := (\Gamma(V(\pi), x(\theta); \zeta_\infty, \mu^*))^{-1}$, where $\mu^*$ is a fixed reward vector in $\mathbb{R}^2$, $\zeta_\infty = \operatorname{argmax}_{x \in \mathcal{X}} \mu^* \cdot x$, $\epsilon = 10^{-2}$. We compare $r(x(\theta))$ with a sampling rule obtained by our algorithm and that of MVR (Vakili et al., 2021).

its decay rate is given as $\lim_{t \to \infty} \left(\Gamma^*(V(\overline{\pi}_t); \zeta_\infty, f)\right)^{-1}$, assuming the limit exists. We note that similar results are well-known in the finite-armed setting (Li et al., 2024, Theorem 2.1), (Russo, 2016, Theorem 1). Lemma 4.1 generalizes these results to general continuous arm sets and general recommendation rules $(\zeta_t)_{t \geq 1}$. In the proof, since we cannot compute $P_t(\zeta_t \notin \mathcal{X}^*(\epsilon))$ directly, we rewrite it using $\sup_{\xi \in \mathcal{X}} P_t(f(\zeta_t) \leq f(\xi) - \epsilon)$, and we further rewrite it to a value of the function $\Gamma^*$.

**Optimization Objective.** Lemma 4.1 indicates that if $(\pi_t)_{t \geq 1}$ is an asymptotically optimal sampling rule and $\overline{\pi}_t$ converges to $\overline{\pi}_\infty \in \mathcal{P}(\mathcal{X})$, then $\overline{\pi}_\infty$ should be a solution of the following optimization problem:

$$\tau_{\mathcal{X}}^*(f; \zeta_\infty) := \inf_{\pi \in \mathcal{P}(\mathcal{X})} \Gamma^*(V(\pi); \zeta_\infty, f) \quad (4)$$
$$= \inf_{\pi \in \mathcal{P}(\mathcal{X})} \sup_{\xi \in \mathcal{X}} \frac{\|\zeta_\infty - \xi\|_{V(\pi)^{-1}}^2}{(\epsilon + f(\zeta_\infty) - f(\xi))^2}.$$

The optimization problem (4) is challenging to solve since the space $\mathcal{P}(\mathcal{X})$ is infinite dimensional and $\Gamma^*(V(\pi); \zeta_\infty, f)$ is non-smooth due to the supremum for $\xi \in \mathcal{X}$. In Figure 1, we show numerical examples of $(\Gamma(V(\pi), \xi; \zeta_\infty, f))^{-1}$ for two sampling rules (ours and MVR). Using the notation in the figure caption, we see that with our sampling rule, the value of $\inf_{\xi \in \mathcal{X}} r(\xi) = (\sup_{\xi \in \mathcal{X}} \Gamma(V(\pi), \xi; \zeta_\infty, \mu^*))^{-1}$ is smaller than that of MVR.

### 4.2. Non-Asymptotic Bound and Stopping Rule

As we discussed in Section 3, $P_t(\zeta_t \notin \mathcal{X}^*(\epsilon))$ is mathematically computable using observations up to round $t$, however, an actual computation is non-trivial especially when the arm set is continuous. The following lemma provides a computable upper bound. For conciseness, we provide a

brief statement here, and a detailed statement in Appendix (Lemma D.1).

**Lemma 4.2** (A computable upper bound of the posterior probability). *Suppose that the assumptions in Lemma 4.1 hold. For any $c > 0$ and for each $t \geq 1$, there exist $b_t \geq 0$ with $b_t = O(\text{poly}(t))$ and $\epsilon_t \geq \epsilon$ with $\lim_{t \to \infty} \epsilon_t = \epsilon$ such that*

$$P_t(\zeta_t \notin \mathcal{X}^*(\epsilon)) \leq a_t + b_t \sup_{\xi \in \mathcal{X}} P_t(f(\zeta_t) \leq f(\xi) - \epsilon_t),$$

*where $a_t = \exp(-t^{1+c})$. Moreover, we have*

$$\limsup_{t \to \infty} \frac{1}{t} \log u_t \leq -\frac{1}{2} \liminf_{t \to \infty} \left( \Gamma^*(V(\overline{\pi}_t); \zeta_\infty, f) \right)^{-1},$$

*where $u_t := a_t + b_t \sup_{\xi \in \mathcal{X}} P_t(f(\zeta_t) \leq f(\xi) - \epsilon_t)$.*

We note that inequalities (2), (3) and Lemma 4.2 imply that $\frac{1}{t} \log u_t$ asymptotically matches $\frac{1}{t} \log P_t(\zeta_t \notin \mathcal{X}^*(\epsilon))$ if $V(\overline{\pi}_t)$ converges, which is the case for our method proposed in Section 5.1. In Lemma D.1, the term $b_t$ is given as (an upper bound) the cardinality of a discretization of $\mathcal{X}$, which can be easily computed since $\mathcal{X} \subset \mathbb{R}^d$ is compact. Moreover, (the proof of) Lemma D.1 implies that

$$\frac{1}{t} \log u_t \leq O\left( \frac{\log t}{t} \right) + \sup_{\xi \in \mathcal{X}} \frac{1}{t} \log P_t(f(\zeta_t) \leq f(\xi) - \epsilon),$$

and the latter term is dominant. By definition and noting the monotonicity of the cumulative distribution function of the standard normal distribution, we can rewrite the above term as follows:

$$\sup_{\xi \in \mathcal{X}} P_t(f(\zeta_t) \leq f(\xi) - \epsilon_t)$$
$$= \Phi\left( -\inf_{\xi \in \mathcal{X}} \frac{\epsilon_t + \mu_t(\zeta_t) - \mu_t(\xi)}{\|\zeta_t - \xi\|_{\Sigma_t^{-1}}} \right),$$

where $\Phi$ is the cumulative distribution function of the standard normal distribution. Therefore, we see that we can compute this term by optimizing a fractional quadratic objective on $\mathcal{X}$, which we detail in Section 5.2.

### 4.3. Choices of Recommendation Rules

A simple example of a recommendation rule is given as the maximizer of the posterior mean, i.e., $\zeta_t = \text{argmax}_{x \in \mathcal{X}} \mu_t(x)$. We call this recommendation rule the greedy recommendation rule. Existing pure exploration methods adopt this recommendation rule (e.g., MVR (Vakili et al., 2021)), and we use it in our experimental benchmarks (Section 7).

Next, we assume that $\Gamma^*(V(\overline{\pi}_t); \zeta_\infty, f)$ converges to $\tau_{\mathcal{X}}^*(f; \zeta_\infty)$ as $t \to \infty$ (as we discuss in Section 6, our sampling rule satisfies this property). Lemma 4.1 implies

that with an optimal recommendation rule $(\zeta_t)_t$, $\zeta_\infty$ should minimize $\tau_{\mathcal{X}}^*(f; \zeta_\infty)$ under the constraint $\zeta_\infty \in \mathcal{X}^*(\epsilon)$. We call such a recommendation rule asymptotically optimal. If an estimation of $\lim_{t \to \infty} \Gamma^*(V(\overline{\pi}_t); \zeta, f)$ (and that of the constraint $\zeta \in \mathcal{X}^*(\epsilon)$) is available for any $\zeta$, one can implement such a recommendation rule by applying any black-box optimization algorithm (such as Bayesian optimization algorithms (Frazier, 2018)). In practice, we can use $\Gamma^*(V(\overline{\pi}_t); \zeta, \mu_t)$ as an estimation of $\Gamma^*(V_\infty; \zeta, f)$ and $\{\zeta \in \mathcal{X} : \mu_t(\zeta) \geq \sup_{x \in \mathcal{X}} \mu_t(x) - \epsilon\}$ as an estimation of the constraint $\{\zeta : \zeta \in \mathcal{X}^*(\epsilon)\}$. We shall discuss algorithms for computing $\Gamma^*(V(\overline{\pi}_t); \zeta, \mu_t)$ in Section 5.2. For a positive-definite matrix $V \in \mathbb{R}^{d \times d}$ and a function $\mu : \mathcal{X} \to \mathbb{R}$, we note that $\text{argmin}_{\zeta \in \mathcal{X}} \Gamma^*(V; \zeta, \mu)$ is equivalent to the furthest answer introduced in the finite-armed setting (Jourdan & Degenne, 2022) and they assume a computation oracle for the furthest answer. similarly, this paper assumes a recommendation rule is given in our theoretical analysis provided in Section 6.

## 5. A Tractable Algorithm for Minimizing the Error Probability

### 5.1. Proposed Method

In Section 3, we formalized the optimization objective Eq. (4) for an optimal sampling rule. In this section, we provide a novel tractable algorithm termed PMCA based on the projected subgradient descent method and discuss its connection to the approximate Carathéodory problem (Combettes & Pokutta, 2023).

**Projected Subgradient Based Method.** The optimization objective (4) is a minimization problem over the space $\mathcal{P}(\mathcal{X})$ of probability measures. One can hope to solve such an optimization problem by Wasserstein gradient flow (Salim et al., 2020). However, an actual implementation of such algorithms is highly non-trivial. More specifically, in the framework of (Salim et al., 2020), an explicit calculation of the proximity operator is unclear for our setting. In this paper, focusing on the simple fact that our objective (4) is a function of a matrix $V(\pi)$, we optimize Eq. (4) over $\mathcal{V}(\mathcal{X})$ by the reparametrization $\pi \mapsto V(\pi)$, where $\mathcal{V}(\mathcal{X}) = \{V(\pi) : \pi \in \mathcal{P}(\mathcal{X})\}$. Moreover, to stabilize the optimization process, for any $\lambda_V \geq 0$, for $\zeta \in \mathcal{X}$, we consider the regularized objective $\inf_{\pi \in \mathcal{P}(\mathcal{X})} \widetilde{\Gamma}^*(V(\pi); \zeta, f)$, where

$$\widetilde{\Gamma}^*(V; \zeta, f) = \Gamma^*(V; \zeta, f) + \lambda_V \|V\|_F^2. \tag{5}$$

We prove that the function $\widetilde{\Gamma}^*(\cdot; \zeta, f)$ is a convex function on $\mathcal{V}(\mathcal{X})$ (Lemma F.2). Consequently, we design our algorithm based on the PSGD (projected subgradient descent method) (cf. (Bubeck, 2015)). For conciseness, we first provide a slightly informal procedure of our method and a difficulty in applying the PSGD. Since the true objective

function is unknown, for each round $t$, we compute a subgradient $g_t$ of an estimation of $\widetilde{\Gamma}^*(\cdot; \zeta, f)$ and compute an approximate projection $\Pi$ of $W_{t+1} = \widetilde{V}_t - \eta_t c_t g_t$ to the space $\mathcal{V}(\mathcal{X})$, where $\widetilde{V}_t \in \mathcal{V}(\mathcal{X})$ is a parameter to be updated by the PSGD based method, $c_t > 0$ is a normalizing factor of the subgradient $g_t$, and $\eta_t$ is a step size.

**Projection and Distribution-Reconstruction.** The reparametrization and PSGD address challenges of the original optimization objective (4), i.e., the infinite dimensionality of $\mathcal{P}(\mathcal{X})$ and the non-smoothness of the objective. However, this approaches introduces new challenges related to the projection onto $\mathcal{V}(\mathcal{X})$ and reconstruction of the sampling distribution from the reparametrization. Noting that the convex closure of $\{xx^\top : x \in \mathcal{X}\}$ is dense in $\mathcal{V}(\mathcal{X})$, if $W \in \mathcal{V}(\mathcal{X})$, then the problem of finding $\pi \in \mathcal{P}(\mathcal{X})$ such that $V(\pi)$ is close to $W$ is equivalent to the approximate Carathéodory problem (Combettes & Pokutta, 2023), i.e., a problem to seek a convex combination close to a given point in a convex set. Therefore, similar to the approximate Carathéodory problem, we compute an approximate projection by the Frank-Wolfe algorithm detailed in Algorithm 2. For each iteration and a matrix $W' \in \mathbb{R}^{d \times d}$, Algorithm 2 solves a linear optimization problem $\operatorname{argmax}_{X \in \mathcal{V}(\mathcal{X})} \operatorname{Tr} W' X$ on $\mathcal{V}(\mathcal{X})$, which is equivalent the quadratic optimization objective $\operatorname{argmax}_{x \in \mathcal{X}} x^\top W' x$ on $\mathcal{X}$. This algorithm solves the aforementioned two challenges (i.e., projection and distribution-reconstruction) simultaneously, i.e., it returns a distribution $\widetilde{\pi}^{(n)}$ such that $V(\widetilde{\pi}^{(n)})$ is an approximate projection of an input $W$ onto $\mathcal{V}(\mathcal{X})$ (Lemma F.4).

**PMCA.** We display pseudo code in Algorithm 1 of our proposed method termed PMCA (a method for posterior error Probability Minimization for Continuous Arm sets). Each round $t$, we compute a subgradient $g_t$ of the function $V \mapsto \widetilde{\Gamma}^*((1 - t^{-\alpha})V + t^{-\alpha}V(\pi_{\exp}); \zeta_t, \mu_t)$, where $\alpha > 0$ is a parameter of the algorithm and we consider a base exploration distribution $\pi_{\exp} \in \mathcal{P}(\mathcal{X})$ with $V(\pi_{\exp}) \succ 0$ to bound the subgradient. With notations defined in the pseudo-code, a subgradient $g_t$ can be computed as follows:

$$g_t := -(1 - t^{-\alpha})(\epsilon + \mu_t(\zeta_t) - \mu_t(\xi_t))^{-2} v_t v_t^\top + 2\lambda_V (1 - t^{-\alpha})\widetilde{V}_t. \tag{6}$$

We denote by $\widetilde{V}_t \in \mathcal{V}(\mathcal{X})$ the parameter updated by PMCA. As we explained above, we compute an approximate projection $V(\widetilde{\pi}_{t+1})$ of $W_{t+1} := \widetilde{V}_t - \eta_t c_t g_t$ to $\mathcal{V}(\mathcal{X})$ by applying Algorithm 2. Then, we define a sampling rule by the mixture distribution $\pi_{t+1} := (1 - (t+1)^{-\alpha})\widetilde{\pi}_{t+1} + (t+1)^{-\alpha}\pi_{\exp}$.

### 5.2. Computation Oracles

For later analysis of the proposed method, we introduce computational oracles for quadratic and quadratic fractional

---

**Algorithm 1** PMCA: a tractable algorithm for error probability minimization

1: **Input**: recommendation rule $(\zeta_t)_{t \geq 1}$, base exploration distribution $\pi_{\exp} \in \mathcal{P}(\mathcal{X})$, step sizes for the PSGD $(\eta_t)_{t \geq 1}$, normalizing factors $(c_t)_{t \geq 1}$, decay rate for the mixture coefficient $\alpha > 0$, step sizes for projection $(n_t)_t$, regularizer $\lambda_V \geq 0$,
2: Initialize: $\pi_1 = \pi_{\exp}, \widetilde{V}_1 = V(\pi_1)$.
3: **for** $t = 1, 2, \ldots,$ **do**
4:     Play $x_t \sim \pi_t$ and observe a noisy reward $y_t$
5:     **// Reparametrization $\pi_t \mapsto V_t$.**
6:     $V_t = (1 - t^{-\alpha})\widetilde{V}_t + t^{-\alpha}V(\pi_{\exp})$
7:     **// Computation of a subgradient $g_t$.**
8:     $\xi_t = \operatorname{argmax}_{\xi \in \mathcal{X}} \Gamma(V_t, \xi; \zeta_t, \mu_t)$.
9:     $v_t := (V_t)^{-1}(\zeta_t - \xi_t)$.
10:    Define $g_t$ by Eq. (6).
11:    **// Update in the matrix space.**
12:    $W_{t+1} = \widetilde{V}_t - \eta_t c_t g_t$.
13:    **// Computation of an approx. projection and distribution-reconstruction.**
14:    $\widetilde{\pi}_{t+1} \leftarrow$ Algorithm 2 with $W = W_{t+1}, n = n_t, \widetilde{\pi}^{(0)} = \widetilde{\pi}_t$
15:    $\pi_{t+1} = (1 - (t+1)^{-\alpha})\widetilde{\pi}_{t+1} + (t+1)^{-\alpha}\pi_{\exp}, \widetilde{V}_{t+1} = V(\widetilde{\pi}_{t+1})$.
16: **end for**

---

**Algorithm 2** Approximate Projection by the Frank-Wolfe Algorithm

**Input**: $W \in \mathbb{R}^{d \times d}, n \geq 1, \widetilde{\pi}^{(0)} \in \mathcal{P}(\mathcal{X}), (\gamma_i)_{i \geq 1}$
**for** $i = 1, 2, \ldots, n$ **do**
    $\widetilde{V}^{(i-1)} = V(\widetilde{\pi}^{(i-1)})$
    $a_i = \operatorname{argmax}_{x \in \mathcal{X}} x^\top (W - \widetilde{V}^{(i-1)})x$
    $\widetilde{\pi}^{(i)} = (1 - \gamma_i)\widetilde{\pi}^{(i-1)} + \gamma_i \delta(a_i)$
**end for**
Output $\widetilde{\pi}^{(n)}$

---

objectives on $\mathcal{X}$.

**Quadratic Objective.** For any symmetric matrix $V \in \mathbb{R}^{d \times d}$ (which is not necessarily definite), we assume that an argmax oracle $\mathcal{O}_Q(V; \mathcal{X})$ for the quadratic objective $x \mapsto x^\top V x$, i.e., $\mathcal{O}_Q(V; \mathcal{X})$ returns a point in $\operatorname{argmax}_{x \in \mathcal{X}} x^\top V x$. Such a computation oracle is assumed in existing work. For instance, Vakili et al. (2021) assumes an argmax oracle for the objective $x \to x^T \Sigma_t^{-1} x$, which is not concave. If $\mathcal{X}$ is defined as linear and quadratic constraints, such an optimization problem is called non-convex quadratic programming. Some problems are known to be solvable in polynomial time, while others are NP-hard (Park & Boyd, 2017).

For example if the arm set $\mathcal{X}$ is defined as an interval constraint $l \leq q(x) \leq u$ with a quadratic function $q$ (this

includes the case of the unit sphere), the quadratic objective can be solved in polynomial time by the Lagrange relaxation (Park & Boyd, 2017). The relaxed problem is a semi-definite problem (convex problem). If we use an interior point method (barrier method), to obtain an $\epsilon'$-optimal solution, the outer loop needs $O(\log(1/\epsilon'))$ iterations and the convergence rate of the inner loop is linear convergence. Per iteration computational complexity is $O(d^4)$ with $O(d^2)$ space complexity (Boyd & Vandenberghe, 2004, Chapter 11.8.3). In general, the problem can be NP-hard. We note that MVR (Vakili et al., 2021) has the same problem and similar to the MVR implementation (PosteriorStandardDeviation) in the BoTorch library (Balandat et al., 2020), we use a non-linear solver (such as L-BFGS-B) with a randomly selected initial point in our experiment.

**Quadratic Fractional Objective.** Next, for $V_1, V_2 \in \mathbb{R}^d$ with $V_2 \succ 0$, we introduce a computation oracle $\mathcal{O}_{\mathrm{QF}}(V_1, V_2; \mathcal{X})$ for quadratic fractional objectives. We assume that $\mathcal{O}_{\mathrm{QF}}(V_1, V_2; \mathcal{X})$ returns a point in $\operatorname{argmax}_{x \in \mathcal{X}}(x^\top V_1 x)/(x^\top V_2 x)$. This is known as fractional programming, and it is well-studied (Dinkelbach, 1967; Shen & Yu, 2018; Schaible, 1976). For instance, Dinkelbach's algorithm iteratively solves $x^{(n)} = \operatorname{argmax}_{x \in \mathcal{X}} A(x) - q_n B(x)$, where $A(x) = x^\top V_1 x$ and $B(x) = x^\top V_2 x$ are the numerator and denominator of the fractional objective, respectively and $q_{n+1} = A(x^{(n)})/B(x^{(n)})$. It is proved that $q_n$ converges to an optimal value $q^*$ and the convergence rate is superlinear (the error goes to zero faster than any geometric sequence) (Schaible, 1976). Then, one can obtain a solution of the original problem by $\operatorname{argmax}_{x \in \mathcal{X}} A(x) - q^* B(x)$. Due to the superlinear convergence rate, we use a limited number of iterations in our experiments.

### 5.3. Tractability of the Proposed Method

We discuss the tractability of PMCA (or computational complexity in terms of the number of oracle calls). In each round $t$, PMCA solves the quadratic fractional objective $\xi_t = \operatorname{argmax}_{\xi \in \mathcal{X}} \Gamma(V_t, \xi; \zeta_t, \mu_t)$ at line 1. This line calls $\mathcal{O}_{\mathrm{QF}}$ once, where $\mathcal{O}_{\mathrm{QF}}$ is the computation oracle for quadratic fractional objectives discussed in Section 5.2. At line 1, PMCA computes an approximate projection by calling Algorithm 2 and it calls the computation oracle $\mathcal{O}_{\mathrm{Q}}$ for quadratic objectives for $n_t$ times. In the theoretical analysis in Section 6, we assume that $\sum_{t=1}^{\infty} n_t^{-1}$ converges. For instance, any $u > 1$, $n_t = t^u$ satisfies this condition. Thus, with this choice of $n_t$, for each round $t$, the computation oracle $\mathcal{O}_{\mathrm{Q}}$ is called for $O(t^u)$ times. For each round $t \geq 1$, we store the support of the discrete probability measure $\widetilde{\pi}_t$ computed by Algorithm 2. Therefore, the space complexity up to round $t$ is given as $O(d^2 + \sum_{s=1}^{t} n_s)$.

## 6. Instance-Dependent (or Asymptotic) Optimality of the Proposed Method

This section provides a theoretical analysis of the proposed method (Algorithm 1). In Section 6.1, we introduce our main theorem (Theorem 6.2) that assures the asymptotic optimality of the proposed method. We derive the main theorem by Lemma 4.1 and convergence analysis of the algorithm, which we introduce in Section 6.2. Moreover, we provide sufficient conditions for satisfying a non-trivial assumption considered in the main results in Section 6.3.

### 6.1. Main Result

We make the following assumptions of the algorithm.

**Assumption 6.1.** We assume that we have access to computational oracles $\mathcal{O}_{\mathrm{Q}}, \mathcal{O}_{\mathrm{QF}}$ for quadratic and quadratic fractional objectives on $\mathcal{X}$ discussed in Section 5.2. Moreover, we assume that the following convergence rate of the recommendation rule $\zeta_\infty = \lim_{t \to \infty} \zeta_t$, a.s.: for any $\delta \in (0,1)$, there exists $c_{\mathcal{X}, f}(\delta) > 0$ and constant $\nu > 0$, such that the following inequality holds for any $t \geq 1$ with probability at least $1 - \delta$:

$$\|\zeta_\infty - \zeta_t\| \leq c_{\mathcal{X}, f}(\delta) t^{-\nu}. \tag{7}$$

We also assume $V(\pi_{\exp}) \succ 0$ and the objective Eq. (5) is minimized at $V \in \mathcal{V}(\mathcal{X})$ with $V \succ 0$. For inputs of Algorithm 1, we assume that $0 < \alpha < \nu$, take $n_t$ so that $\sum_{t=1}^{\infty}(n_t + 2)^{-1} \leq 1$ (e.g., $n_t \approx t^s$ with $s > 1$). We take an input of Algorithm 2 as $\gamma_i = 2/(i+1)$.

Here, the most non-trivial assumption is that for the convergence rate of the recommendation rule Eq. (7). We note that because $\zeta_t$ depends on the observation history $(x_1, y_1), \ldots, (x_t, y_t)$, $\nu$ can depend on other parameters of the algorithm. In Section 6.3, we provide examples satisfying the assumption Eq. (7) in the case when $\zeta_t = \operatorname{argmax} \mu_t$. Under these assumptions, the main result of this paper is given as follows.

**Theorem 6.2** (Instance-Dependent Optimality). *Suppose Assumption 6.1 hold, and let $(\pi_t)_{t \geq 1}$ be the sampling rule of Algorithm 1, and $(\zeta_t)_{t \geq 1}$ be a recommendation rule with $\zeta_\infty = \lim_{t \to \infty} \zeta_t$ a.s. Furthermore, we assume that $0 < \alpha < \min(\nu, \frac{1}{8})$ and take $\eta_t = t^{-2\alpha - 1/2}$, $c_t = 1, \lambda_V = 0$. Then, the following holds:*

$$\lim_{t \to \infty} \frac{1}{t} \log P_t(\zeta_t \notin \mathcal{X}^*(\epsilon)) = -\frac{1}{2\tau_{\mathcal{X}}^*(f; \zeta_\infty)} \tag{8}$$

This theorem implies that the decay rate of the error probability of our sampling rule is given as $(2(\tau_{\mathcal{X}}^*(f; \zeta_\infty)))^{-1}$, and Lemma 4.1 and (4) imply that it is equal to the optimal rate.

## 6.2. Convergence Analysis

Theorem 6.2 can be proved by Lemma 4.1 and the results on the convergence analysis introduced below. Since we consider Gaussian posterior distributions, Lemma 4.1 requires two conditions, i.e., $\lim_{t\to\infty} \mu_t(x) = f(x)$ and $V(\overline{\pi}_t)$ converges to an optimal matrix of our objective (4). The former condition regarding $\mu_t$ is a weak requirement and satisfied by any sampling policy with $\liminf_{t\to\infty} \lambda_{\min}(V(\pi_t)) > 0$. The latter condition is more subtle and to satisfy the condition, we need to solve the optimization problem (5). Therefore, we can deduce Theorem 6.2 by the following proposition:

**Proposition 6.3.** *Suppose the same assumptions as in Theorem 6.2 hold. For the sampling rule $(\pi_t)_{t\geq 1}$ of PMCA, we have the following:*

$$\lim_{t\to\infty} \Gamma^*(V(\overline{\pi}_t); \zeta_\infty, f) = \tau_{\mathcal{X}}^*(f; \zeta_\infty).$$

In Proposition 6.3, we assume the step sizes $\eta_t$ is given as $t^{-2\alpha-1/2}$. We shall show that a converge result can be proved with more general (standard) choices of step sizes in Proposition D.3.

## 6.3. Sufficient Condition for the Assumption Eq. (7)

In subsection, we provide examples of $\mathcal{X}$ satisfying the assumption Eq. (7) in the case of the greedy recommendation rule $\zeta_t = \arg\max \mu_t$. First, we introduce the following sufficient condition for Eq. (7).

**Proposition 6.4** (Sufficient Condition for Eq. (7)). *We assume that for any vector $\mu \in \mathbb{R}^d$ with $\mu \neq 0$ $\arg\max_{x\in\mathcal{X}} \mu \cdot x$ is singleton (an optimal arm is unique) and denote $\zeta_\mu^* = \arg\max_{x\in\mathcal{X}} \mu \cdot x$. We further assume that for any $\mu$ with $\mu \neq 0$ there exist constants $c_{\mathcal{X},\mu}', \kappa_{\mathcal{X}} > 0$ such that the following inequality holds :*

$$\|x - \zeta_\mu^*\|^{\kappa_{\mathcal{X}}} \leq c_{\mathcal{X},\mu}' \left| \mu \cdot \zeta_\mu^* - \mu \cdot x \right|. \tag{9}$$

*Let $\zeta_t = \arg\max \mu_t$ the greedy recommendation rule and $(x_t)_{t\geq 1}$ be a sampling rule with $\lambda_{\min}(V_t) \gtrsim t^{-\alpha}$. Then, the inequality (7) holds with $\nu = \frac{1-\alpha}{2\kappa_{\mathcal{X}}}$.*

Since the condition $\lambda_{\min}(V_t) \gtrsim t^{-\alpha}$ holds for the sampling rule of Algorithm 1, if the inequality (9) is satisfied, then the assumption (7) is satisfied. We also note that the condition $\alpha < \min(1/3, \nu)$ is satisfied if $\alpha < \min(1/3, 1/(2\kappa_{\mathcal{X}}+1))$ in the setting of Proposition 6.4.

It is easy to see that the unit sphere $\mathcal{X} = \{x \in \mathbb{R}^d : \|x\| = 1\}$ satisfies the condition (9). More generally, we can show that if $\mathcal{X}$ is a hypersurface with a "positive curvature", then $\mathcal{X}$ satisfies the condition.

**Proposition 6.5** (Examples of $\mathcal{X}$ satisfying Eq. (7)). *We assume that $\zeta_\mu^* := \arg\max_{x\in\mathcal{X}} \mu \cdot x$ is unique for any $\mu \in \mathbb{R}^d$ with $\|\mu\| \neq 0$. If for any $x_0 \in \mathcal{X}$ there exists a neighborhood $\mathcal{U}_{x_0} \subset \mathbb{R}^d$ of $x_0$ and an analytic function $\phi_{x_0} : \mathcal{U}_{x_0} \to \mathbb{R}$ such that $\mathcal{X} \cap \mathcal{U}_{x_0}$ is given as $\{x \in \mathcal{U}_{x_0} : \phi_{x_0}(x) = 0\}$, $\|\nabla\phi_{x_0}(x)\|$ is bounded on $\mathcal{U}_{x_0}$, and the minimal eigenvalue of the Hessian of $\phi_{x_0}$ at $x_0$ is larger than a positive constant $c > 0$ independent of $x_0$, then $\mathcal{X}$ satisfies the condition (9) with $\kappa_{\mathcal{X}} = 2$.*

We note that an existing work also assumed arm sets with positive curvature (Banerjee et al., 2023, Definition 2.1).

# 7. Experiments

We coduct experiments in simple synthetic environments for the case $\mathcal{X} \subset \mathbb{R}^2$, and empirically demonstrate that our proposed method outperforms existing pure exploration algorithms for continuous arm sets.

**Motivation.** Jedra & Proutiere (2020) proposed a pure exploration algorithm in the case when the arm set $\mathcal{X}$ is the unit sphere, and they showed that an optimal sampling policy is the round-robin manner using the orthonormal basis. Due to the symmetry of the arm set (unit sphere), this uniform exploration distribution is in fact optimal; however, if the arm set is a more general set, then this distribution may not be optimal. Therefore, we compare our algorithm with existing methods in several (synthetic) problem instances with $\mathcal{X} \subset \mathbb{R}^2$.

**Setup of the experiments.** In this simple setting, $\mathcal{X}$ is defined as a subset $\{(\cos(\theta), \sin(\theta)) : \theta \in [\theta_0, \theta_1]\}$ of the unit circle in $\mathbb{R}^2$, and $f(x) = (\cos(\theta_f), \sin(\theta_f)) \cdot x$, $\lambda = 10^{-2}, \epsilon = 10^{-2}$. Here, for simplicity, we consider a deterministic function $f$ and we understand that $f$ is sampled before the round starts. Specifically, in this section, we consider the case when $\theta_f = a\pi, \theta_0 = 0, \theta_1 = b\pi$. We have conducted experiments for $a = 0.0, 0.2, \ldots, 1.8$ and $b = 0.0, 0.2, \ldots, 1.8$ with $a < b$. In this section, we only show the case when $a = 0.2$, $b = 0.4, 0.6, 0.8, 1.0$. We refer to Appendix G for further experimental results.

**Evaluation metric.** For all Sampling rules, we use the same recommendation rule defined as $\zeta_t = \arg\max_{x\in\mathcal{X}} \mu_t(x)$. For a sampling rule $(\pi_t)_{t\geq 1}$, we let $p_t := \sup_{\xi\in\mathcal{X}} P_t(f(\zeta_t) \leq f(\xi) - \epsilon)$ and use $p_t$ as an evaluation metric (the lower is the better). By Lemma 4.2, assuming $\epsilon_t \approx \epsilon$, this corresponds to the computational upper bound of $P_t(\zeta_t \notin \mathcal{X}^*(\epsilon))$ since $a_t$ and $b_t$ are independent of sampling rules.

**Baselines.** We compare our method PMCA to the uniform random policy and MVR (Vakili et al., 2021), where the uniform random policy is a strong baseline in a pure exploration problem (as we discussed above, Jedra & Proutiere (2020) proved a uniform exploration is optimal in the case of the unit sphere). MVR is an optimal method for any arm

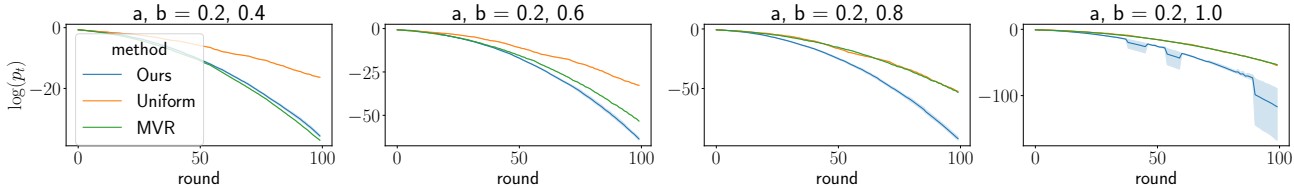

Figure 2. Comparison of our method to existing pure exploration methods in several problem instances.

sets for a pure exploration problem (simple regret minimization), however, as we discussed in the introduction, their method is instance-independent.

**Results.** Figure 2 shows the experimental results for several problem instances, where the error bands represent 95%-confidence interval over 10 repetition. MVR outperforms Uniform, but, is sub-optimal for some problem instances.

We provide more details of the experiments, experimental results, and CPU time for each method in Appendix G.

## 8. Concluding Remarks

### 8.1. Conclusion

We proposed a pure exploration algorithm for general continuous arm sets that is tractable and achieves an instance-dependent optimality. Moreover, we empirically show that our method outperforms baselines in terms of the computational upper bound of the non-asymptotic error probability.

### 8.2. Limitations and Future Work

A limitation of this study is the asymptotic nature of the optimality result. Developing a finite-time analysis for a practical algorithm would be an important direction for future research. In addition, we focus on the optimization of a sampling rule and assume a recommendation rule is given. As discussed in Section 4.3, an (asymptotically) optimal method requires both optimal sampling and an optimal recommendation rule and we only discuss an optimal sampling rule in this study. Moreover, some of our theoretical results (those provided in Section 6.3) and experimental results focus only on the greedy recommendation rule.

A generalization to the Gaussian process reward model would be an important future work since it would enable the development of a practical Bayesian optimization algorithm with instance-dependent optimality. However, existing analysis (Vakili et al., 2021) indicates that exponential decay of the error probability may not be guaranteed depending on kernel smoothness, necessitating new theoretical analysis. Therefore, we leave it as a future work.

## Impact Statement

This paper presents work whose goal is to advance the field of Machine Learning, especially for the area of the online optimization with bandit feedback. There are many potential societal consequences of our work, none which we feel must be specifically highlighted here.

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

*Table 1.* List of Notations

| Symbol | Description |
|---|---|
| $\mathcal{X}$ | the arm set |
| $f$ | the reward function |
| $f_t$ | $f$ conditioned on $\mathcal{F}_t$ |
| $\mathcal{P}(\mathcal{X})$ | the space of probability measures on $\mathcal{X}$ |
| $(\zeta_t)_{t \geq 1}$ | recommendation rule |
| $\zeta_\infty$ | $\lim_{t \to \infty} \zeta_t$ (a.s. convergence) |
| $(\pi_t)_{t \geq 1}$ | sampling rule |
| $(\boldsymbol{\omega}_t)_{t \geq 1}$ | the noise stochastic process |
| $\mathcal{V}(\mathcal{X})$ | $\{V(\pi) : \pi \in \mathcal{P}(\mathcal{X})\}$ |
| $\mu_t, \Sigma_t$ | the posterior mean and covariance matrix |
| $\Gamma^*(V(\pi); \zeta, f)$ | the objective function for an optimal allocation $\pi$ (Eq. (1)) |
| $\widetilde{\Gamma}^*(V(\pi); \zeta, f)$ | the regularized objective with a regularizer $\lambda_V$ (Eq. (5)) |
| $\Gamma(V, \xi; \zeta, \mu)$ | $\frac{\|\zeta - \xi\|_{V^{-1}}^2}{(\epsilon + \mu(\zeta) - \mu(\xi))^2}$ |
| $P_t$ | a posterior probability distributions of random rewards $f_t$ |
| $\overline{\pi}_t$ | $\frac{1}{t} \sum_{s=1}^{t} \pi_t$ |
| $\|V\|_\mathrm{F}$ | Frobenius norm $\sqrt{\mathrm{Tr}\, VV^\top}$ of a matrix $V$ |
| $\mathcal{S}^d$ | the space of symmetric matrices in $\mathbb{R}^{d \times d}$ |
| $\delta(a)$ | Dirac delta supported on a singleton $\{a\}$, where $a \in \mathcal{X}$ |
| $\Phi$ | the cumulative distribution function of the standard normal distribution |
| $V(\pi)$ | $\mathbb{E}_{x \sim \pi}\left[xx^\top\right]$ |

## A. Notation Table

We provide the list of notations in Table 1.

## B. Another Interpretation of the Objective

To provide another interpretation of $\tau_{\mathcal{X}}^*(f; \zeta)$, we introduce known results on characteristic time (Jourdan & Degenne, 2022) for the BAI problem in the case of the finite-armed setting. In this subsection, we consider a non-random linear reward function $f : \mathcal{X} \to \mathbb{R}$ and a finite discretization $\widehat{\mathcal{X}}$ of $\mathcal{X}$. For simplicity, we assume $\mathrm{argmax}_{x \in \widehat{\mathcal{X}}} f(x)$ is unique. Let $\mathcal{A}$ be a $(\epsilon, \delta)$-PAC strategy consisting of sampling rule for selecting arms $x_t \in \widehat{\mathcal{X}}$ for each round $t$, and recommendation rule $\hat{z} \in \widehat{\mathcal{X}}$, and stopping time $\tau_\delta$ such that $P(\tau_\delta < \infty, \hat{z} \notin \widehat{\mathcal{X}}^*(\epsilon)) \leq \delta$. An $(\epsilon, \delta)$-PAC strategy is called asymptotically greedy if $\lim_{\delta \to 0} P(\tau_\delta < \infty, \hat{z} \neq \mathrm{argmax}_{x \in \widehat{\mathcal{X}}} f(x)) = 0$.

**Proposition B.1** ((Jourdan & Degenne, 2022), Theorem 2.1, Lemma 2.2, Lemma C.1). *(i) For any asymptotically greedy $(\epsilon, \delta)$-PAC strategy with stopping time $\tau_\delta$, we have*

$$\liminf_{\delta \to 0} \frac{\mathbb{E}[\tau_\delta]}{\log 1/\delta} \geq \inf_{\pi \in \mathcal{P}(\widehat{\mathcal{X}})} \sup_{\xi \in \widehat{\mathcal{X}}} 2 \frac{\|\zeta^* - \xi\|_{V^\dagger(\pi)}^2}{(\epsilon + f(\zeta^*) - f(\xi))^2}. \tag{10}$$

*Here, $\zeta^* = \mathrm{argmax}_{x \in \widehat{\mathcal{X}}} f(x)$ and $V^\dagger(\pi)$ denotes the Moore-Penrose pseudo inverse of $V(\pi)$. (ii) For any $(\epsilon, \delta)$-PAC strategy with stopping time $\tau_\delta$, we have*

$$\liminf_{\delta \to 0} \frac{\mathbb{E}[\tau_\delta]}{\log 1/\delta} \geq \inf_{\zeta \in \widehat{\mathcal{X}}^*(\epsilon)} \inf_{\pi \in \mathcal{P}(\widehat{\mathcal{X}})} \sup_{\xi \in \widehat{\mathcal{X}}} 2 \frac{\|\zeta - \xi\|_{V^\dagger(\pi)}^2}{(\epsilon + f(\zeta) - f(\xi))^2}. \tag{11}$$

The LHS of Eq. (11) (resp. Eq. (10)) is called characteristic time (resp. greedy characteristic time) and any optimal strategy should minimize characteristic time. For instance, Lazy Track-and-Stop (Jedra & Proutiere, 2020) minimizes characteristic time over the probability distribution on a finite arm sets for $f(x) = \mu_t(x)$, where $\mu_t$ is an estimation of the reward function. L$\epsilon$BAI (Jourdan & Degenne, 2022) requires a computation oracle for the optimization problem (11) that returns an optimal $\zeta$ and $\pi$ for $f(x) = \mu_t(x)$.

We note that one can take any fine discretization $\widehat{\mathcal{X}}$ of $\mathcal{X}$ in Proposition B.1. However, taking a fine discretization could hurt the performance of an algorithm. If $\widehat{\mathcal{X}}$ is a sufficiently fine discretization of $\mathcal{X}$ so that we can identify $\widehat{\mathcal{X}}$ with the continuous arm set $\mathcal{X}$ and if $V(\pi^*)$ is positive definite for any optimal allocation $\pi^*$, then we can apply our algorithm to the characteristic time minimization in the problem setting in Proposition B.1.

Intuitively, an asymptotically greedy strategy corresponds to a recommendation rule $(\zeta_t)_{t \geq 1}$ satisfying $\lim_{t \to \infty} \zeta_t = \arg\max_{x \in \mathcal{X}} f(x)$ in our setting. In this case, our objective $\tau_{\mathcal{X}}^*(f; \zeta)$ corresponds to the minimization of greedy characteristic time (10). Next, we assume that a recommendation rule $(\zeta_t)_{t \geq 1}$ satisfies $\lim_{t \to \infty} \zeta_t \in \arg\min_{\zeta \in \mathcal{X}} \tau_{\mathcal{X}}^*(f; \zeta)$. One could construct such a recommendation rule if we have an estimation of $\tau_{\mathcal{X}}^*(f; \zeta)$ for $\zeta \in \mathcal{X}$ in each time step $t$, and apply any black-box optimization methods (such as Bayesian optimization) to the estimated objective for $\zeta$. Then, the objective $\tau_{\mathcal{X}}^*(f; \zeta)$ corresponds to the minimization of characteristic time (11).

## C. Related Work (Extended)

This paper studies a pure exploration problem with linear bandit feedback on a *continuous* arm set $\mathcal{X} \subset \mathbb{R}^d$ and our objective is to minimize the posterior probability $P_t(\zeta_t \notin \mathcal{X}^*(\epsilon))$ of misidentification under a Bayesian linear reward model. As we mentioned in the introduction, Bhat & Amballa (2022) studied a $(\epsilon, \delta)$-PAC learning problem with linear bandit feedback on a continuous arm set, where the objective is to minimize the stopping time of an algorithm that returns an $\epsilon$-optimal arm with confidence $1 - \delta$. The $(\epsilon, \delta)$-PAC learning problem with linear bandit feedback has been well-studied especially for the finite-armed setting (Jourdan & Degenne, 2022; Jedra & Proutiere, 2020; Soare et al., 2014; Xu et al., 2018; Fiez et al., 2019; Degenne et al., 2019; Jedra & Proutiere, 2020; Mohammadi Zaki & Gopalan, 2022). Specifically, Jourdan & Degenne (2022) studied the transductive linear bandits (a more general setting than linear bandits), proposed an $(\epsilon, \delta)$-PAC learning algorithm, and discussed asymptotic optimality as $\delta \to 0$ (i.e., a limit to the case of 100% confidence). Although our objective is to minimize the posterior probability $P_t(\zeta_t \notin \mathcal{X}^*(\epsilon))$, our theoretical analysis is parallel to that of the $(\epsilon, \delta)$-PAC learning setting, i.e., we discuss asymptotic optimality of our algorithm as $t \to \infty$ (Lemma 4.1, Theorem 6.2) and discuss a stopping rule that ensures the posterior probability of misidentification is less than $\delta$, where $\delta \in (0, 1)$ is given. We also note that an asymptotic analysis of the posterior probability of misidentification is standard in the finite-armed setting (Li et al., 2024; Russo, 2016).

Despite extensive existing research for pure exploration problems with linear bandit feedback, most papers focus on the case of finite-armed settings due to the difficulties mentioned in the introduction. To the best of our knowledge, it is an open problem to develop a trackable algorithm for a pure exploration problem on a general continuous arm set with an *instance-dependent* optimality. Other than linear bandit literature, in general (not necessarily linear) Gaussian process bandits, (Vakili et al., 2021) proposed an optimal algorithm pure exploration (simple regret minimization). However, their analysis is also instance-independent.

In the finite and discrete armed setting (i.e., there is no structural reward model), the BAI problem has been extensively studied. Since the objective introduced in Section 3 is closely related to the characteristic time in the fixed confidence setting (Jourdan & Degenne, 2022; Jedra & Proutiere, 2020), we mainly restrict our attention to the setting. In the finite and discrete armed setting, Glynn & Juneja (2004) derived a condition for an asymptotically optimal allocation. More recent papers provided more fine-grained analysis and optimal algorithms; Track-and-Stop (Garivier & Kaufmann, 2016), Frank-Wolfe based method (Wang et al., 2021), Top-Two algorithms (Russo, 2016; Qin et al., 2017; Shang et al., 2020; Jourdan et al., 2022; You et al., 2023). Track-and-Stop and the Frank-Wolf-based method iteratively optimize a sampling distribution. However, unlike their setting, our probability space $\mathcal{P}(\mathcal{X})$ is infinite-dimensional, so it is highly non-trivial to generalize these existing results. Moreover, due to the correlation between arms, it is not trivial to generalize some existing algorithms such as top-two algorithms. The BAI problem has been extended to the linear bandit setting. However, most existing works focus on the finite armed setting (Soare et al., 2014; Xu et al., 2018; Fiez et al., 2019; Degenne et al., 2019; Jedra & Proutiere, 2020; Mohammadi Zaki & Gopalan, 2022; Li et al., 2024). Therefore, similarly to the discrete finite case, it is non-trivial to generalize these existing algorithms to the case of continuous arm sets.

## D. Additional or Detailed Theoretical Results

**Lemma D.1** (A computable upper bound of the posterior probability). *For any $h > 0$, let $\widetilde{\mathcal{X}}_h \subset \mathcal{X}$ be a finite subset satisfying the following condition: for ant $x \in \mathcal{X}$, there exists $x' \in \widetilde{\mathcal{X}}_h$ such that $\|x - x'\| \leq h$. For any $\delta' \in (0, 1)$, we take $B_\delta' > 0$ so that $P(d/2, B_\delta'^2/2) \geq 1 - \delta'$, where $P$ is the regularized gamma function (i.e., $x \mapsto P(d/2, x/2)$ is the*

*cumulative distribution function of the chi squared distribution $\chi_d^2$).*

1. *Then, for each $\epsilon > 0$, we let*

$$u_t(\delta', h; \widetilde{\mathcal{X}}_h) = \delta' + |\widetilde{\mathcal{X}}_h| \sup_{\xi \in \mathcal{X}} P_t(f(\zeta_t) \le f(\xi) - \epsilon - hB_{t,\delta'}).$$

   *where $B_{t,\delta'}$ is given as $B_{t,\delta'} = \|\mu_t\| + \|\Sigma_t^{-1/2}\|_F B_{\delta'}$. Then, we have the following non-asymptotic upper bound: $P_t(\zeta_t \notin \mathcal{X}^*(\epsilon)) \le u_t(\delta', h; \widetilde{\mathcal{X}}_h)$.*

2. *Moreover, let $c > 0$ be any positive number. For each $t \ge 1$, if we take take $\delta'_t = \exp(-t^{1+c})$, then there exists a sequence of finte sets $(\mathcal{X}_{h_t})_{t \ge 1}$ with $|\mathcal{X}_{h_t}| = O(\text{poly}(t))$ and $(h_t)_{t \ge 1}$ such that*

$$\limsup_{t \to \infty} \frac{1}{t} \log u_t(\delta'_t, h_t; \mathcal{X}_{h_t})$$
$$\le -\frac{1}{2} \liminf_{t \to \infty} \left(\Gamma^*(V(\overline{\pi}_t); \zeta_\infty, f)\right)^{-1}. \tag{12}$$

*Remark* D.2. Assuming $\mathcal{X} \subset [0,1]^d$, we make the statement of the lemma more explicit. Using notation of 2 of D.1, by the tail probability of chi squared distribution, we can take $h_t = t^{-(1+c)}$ and $|\mathcal{X}_h|$ is given as $(\sqrt{d}h^{-1})^d$. Then, $u_t(\delta'_t, h_t, \mathcal{X}_{h_t})$ gives an explicit upper bound.

**Proposition D.3.** *Suppose Assumption 6.1 hold. Furthermore, we let $\lambda_V = 0$, assume that $0 < \alpha < \min(\nu, \frac{1}{3})$, $c_t > 0$ is given as $c_t = \frac{1}{\max(c, \|g_t\|_F)}$ with a constant $c > 0$, $(\eta_t)_{t \ge 1}$ satisfies $\lim_{t \to \infty} \eta_t = 0$, and $\sum_{t=1}^\infty \eta_t = \infty$. Let $(\pi_t)_{t \ge 1}$ be the sampling rule defined by Algorithm 1. Then, we have the following a.s., $\liminf_{t \to \infty} \Gamma^*(V(\pi_t); \zeta_\infty, f) = \tau^*(f; \zeta_\infty)$.*

## E. Some Measure Theoretic Details

We provide some measure theoretical details omitted in Section 3. We assume that $f$, $(\omega_t)_t$ and randomness of the algorithm are defined on a probability space $(\Omega, \mathcal{F}, P)$ and the probability space is the product of two probability spaces $(\Omega_{\text{gp}}, \mathcal{F}_{\text{gp}}, P_{\text{gp}})$ and $(\Omega_{\text{n}}, \mathcal{F}_{\text{n}}, P_{\text{n}})$, where the Gaussian process $f$ (resp. the i.i.d noise process $(\omega_t)_t$ and randomness of the algorithm) is defined on $(\Omega_{\text{gp}}, \mathcal{F}_{\text{gp}}, P_{\text{gp}})$ (resp. $(\Omega_{\text{n}}, \mathcal{F}_{\text{n}}, P_{\text{n}})$).

## F. Proofs

### F.1. Overview of the Proofs

In Appendix F.2, we provide a proof of Lemmas 4.1 and D.1. In Appendix F.3, we shall see that functions $\Gamma(\cdot; \xi; \zeta, f)$ and $\Gamma^*(\cdot; \zeta, f)$ is convex on $\mathcal{V}(\mathcal{X})$. In Appendix F.4, we introduce some lemmas required for the proofs of Proposition D.3 and Theorem 6.2. More specifically, we introduce results for subgradients, an approximate projection, an error analysis of an estimated objective function of $\Gamma^*(\cdot; \zeta, f)$. Using these lemmas, we prove Proposition D.3 in Appendix F.5, and prove Theorem 6.2 (and Proposition 6.3) in Appendix F.6. Finally, we prove Proposition 6.5 in Appendix F.7.

### F.2. Proofs of Lemmas 4.1 and D.1

In this section, first, we proof the inequality (2), and Lemma D.1, then we prove the inequality (3). For the proof, we introduce the following lemma:

**Lemma F.1.** *Let $(\zeta_t)_{t \ge 1}$ be a recommendation rule. Then, the following holds for any $t \ge 1$:*

$$\sup_{\xi \in \mathcal{X}} P_t\left(f(\zeta_t) \le f(\xi) - \epsilon\right) \le P_t\left(f(\zeta_t) \le \sup_{\xi \in \mathcal{X}} f(\xi) - \epsilon\right).$$

*Proof.* For any $\xi$, we see that the event $f(\zeta_t) \le f(\xi) - \epsilon$ implies $f(\zeta_t) \le \sup_{\xi \in \mathcal{X}} f(\xi) - \epsilon$. Therefore, $\sup_{\xi \in \mathcal{X}} P_t(f(\zeta_t) \le f(\xi) - \epsilon) \le P_t(f(\zeta_t) \le \sup_{\xi \in \mathcal{X}} f(\xi) - \epsilon)$. $\square$

*Proof of* (2). Let $(\overline{\pi}_{t_i})_{i \geq 1}$ be any convergence subsequence of $(\overline{\pi}_t)_{t \geq 1}$ and put $V = \lim_{i \to \infty} V(\overline{\pi}_{t_i})$. Then, the assumption of the lemma, we have $\lambda_{\min}(V) > 0$. To show (2), it is enough to prove

$$\lim_{i \to \infty} \frac{1}{i} \log P_i(\zeta_i \notin \mathcal{X}^*(\epsilon)) \geq -\frac{1}{2} \left(\Gamma^*(V; \zeta_\infty, f)\right)^{-1}.$$

In the rest of the proof, by abuse of notation, we assume $(\overline{\pi}_t)$ converges, $\lim_{t \to \infty} V(\overline{\pi}_t) = V$, and prove the following:

$$\lim_{t \to \infty} \frac{1}{t} \log P_t(\zeta_t \notin \mathcal{X}^*(\epsilon)) \geq -\frac{1}{2} \left(\Gamma^*(V; \zeta_\infty, f)\right)^{-1}. \tag{13}$$

Let $\xi$ is a $\mathcal{F}_t$-measurable random variable. For $t \geq 1$, we let $X_t := \epsilon + f_t(\zeta_t) - f_t(\xi)$. Then conditioned on $\mathcal{F}_t$, $X_t$ follows the normal distribution:

$$X_t \sim \mathcal{N}\left(\epsilon + \mu_t(\zeta_t) - \mu_t(\xi), (\zeta_t - \xi)^\top \Sigma_t^{-1}(\zeta_t - \xi)\right). \tag{14}$$

Thus, we have

$$P_t(f(\zeta_t) \leq f(\xi) - \epsilon) = P(X_t \leq 0) = \Phi\left(-t^{1/2} \frac{\epsilon + \mu_t(\zeta_t) - \mu_t(\xi)}{\|\zeta_t - \xi\|_{(\Sigma_t/t)^{-1}}}\right), \tag{15}$$

where $\Phi$ is the cumulative distribution function of the standard normal distribution. Using the asymptotic formula of the complementary error function for sufficiently large $x > 0$,

$$\mathrm{erfc}(x) = \frac{\exp(-x^2)}{x\sqrt{\pi}} \left(1 + O(x^{-2})\right),$$

we have the following for sufficiently large $x > 0$,

$$\log \Phi(-x) = \log \mathrm{erfc}(x/\sqrt{2}) - \log 2$$
$$= -\frac{x^2}{2} - \log x\sqrt{2\pi} + \log\left(1 + O(x^{-2})\right).$$

We let $r_t := \frac{\epsilon + \mu_t(\zeta_t) - \mu_t(\xi)}{\|\zeta_t - \xi\|_{(\Sigma_t/t)^{-1}}}$, Then, by assumptions, we have $\lim_{t \to \infty} r_t = \frac{\epsilon + f(\zeta) - f(\xi)}{\|\zeta - \xi\|_{V^{-1}}} > 0$. Therefore, by Eq. (15), for sufficiently large $t$, we have the following:

$$\frac{1}{t} \log P_t(f(\zeta_t) \leq f(\xi) - \epsilon) = -\frac{r_t}{2} - \frac{1}{t} \log r_t \sqrt{2\pi t^{1/2}} + \frac{1}{t} \log\left(1 + O(t^{-1} r_t^{-2})\right). \tag{16}$$

Thus, it follows that

$$\lim_{t \to \infty} \frac{1}{t} \log P_t(f(\zeta_t) \leq f(\xi) - \epsilon) = -\frac{1}{2} \frac{(\epsilon + f(\zeta) - f(\xi))^2}{\|\zeta - \xi\|_{V^{-1}}^2}. \tag{17}$$

The above argument holds for any $\xi$ even on the event $\xi = \zeta$ (if $\xi = \zeta$, then $\lim_{t \to \infty} r_t = \infty$ and the both sides of Eq. (16) tend to $-\infty$ and we understand both sides of the equation above represent $-\infty$). Since $\lim_{\xi \to \zeta} \frac{(\epsilon + f(\zeta) - f(\xi))^2}{\|\zeta - \xi\|_{V^{-1}}^2} = \infty$, for any small open neighborhood $\mathcal{U}$ of $\zeta$, we have $\inf_{\xi \in \mathcal{X}} \frac{(\epsilon + f(\zeta) - f(\xi))^2}{\|\zeta - \xi\|_{V^{-1}}^2} = \inf_{\xi \in \mathcal{X} \setminus \mathcal{U}} \frac{(\epsilon + f(\zeta) - f(\xi))^2}{\|\zeta - \xi\|_{V^{-1}}^2}$. Thus, by (17), we have

$$\sup_{\xi \in \mathcal{X} \setminus \mathcal{U}} \lim_{t \to \infty} \frac{1}{t} \log P_t(f(\zeta_t) \leq f(\xi) - \epsilon) = -\frac{1}{2} \left(\Gamma^*(V; \zeta_\infty, f)\right)^{-1}. \tag{18}$$

Since $\zeta_t \to \zeta_\infty$, for sufficiently large $t$, convergence of $t$ in LHS of (18) is uniform w.r.t $\xi \in \mathcal{X} \setminus \mathcal{U}$. Therefore, we can exchange $\lim_{t \to \infty}$ and $\sup_{\xi \in \mathcal{X} \setminus \mathcal{U}}$ and it follows that

$$\lim_{t \to \infty} \frac{1}{t} \sup_{\xi \in \mathcal{X} \setminus \mathcal{U}} \log P_t(f(\zeta_t) \leq f(\xi) - \epsilon) = -\frac{1}{2} \left(\Gamma^*(V; \zeta_\infty, f)\right)^{-1}. \tag{19}$$

By (16) with $r_t = \frac{\epsilon + \mu_t(\zeta_t) - \mu_t(\xi)}{\|\zeta_t - \xi\|_{(\Sigma_t/t)^{-1}}}$ and $\lim_{t \to \infty} r_t = \frac{\epsilon + f(\zeta_\infty) - f(\xi)}{\|\zeta_\infty - \xi\|_{V^{-1}}}$, we can take a neighborhood $\mathcal{U}$ of $\zeta_\infty$ and a sufficiently large $t_0 \geq 1$ such that $\sup_{\xi \in \mathcal{X} \setminus \mathcal{U}} \log P_t(f(\zeta_t) \leq f(\xi) - \epsilon) = \sup_{\xi \in \mathcal{X}} \log P_t(f(\zeta_t) \leq f(\xi) - \epsilon)$ for any $t \geq t_0$, i.e., the supremum is not attained for $\xi \in \mathcal{U}$. Thus, we see that the following holds:

$$\lim_{t \to \infty} \frac{1}{t} \sup_{\xi \in \mathcal{X}} \log P_t(f(\zeta_t) \leq f(\xi) - \epsilon) = -\frac{1}{2} \left(\Gamma^*(V; \zeta_\infty, f)\right)^{-1}. \tag{20}$$

The inequality (13) follows from (20) and Lemma F.1. This completes the proof. $\square$

Next, we provide proofs of Lemma D.1 and the inequality (3).

*Proof of Lemma D.1 and (3).* Let $\widetilde{\mathcal{X}} \subset \mathcal{X}$ be any finite subset and define

$$\mathcal{X}^*(\epsilon; \widetilde{\mathcal{X}}) = \left\{ x \in \mathcal{X} : f(x) > \sup_{\xi \in \widetilde{\mathcal{X}}} f(\xi) - \epsilon \right\}.$$

We define $p_t(\epsilon; \widetilde{\mathcal{X}})$ by

$$p_t(\epsilon; \widetilde{\mathcal{X}}) := P_t\left( \zeta_t \notin \mathcal{X}^*(\epsilon; \widetilde{\mathcal{X}}) \right).$$

Then, by a union bound, we have

$$\begin{aligned}
\log p_t(\epsilon; \widetilde{\mathcal{X}}) = \log P_t & \left( \bigcup_{\xi \in \widetilde{\mathcal{X}}} \{ f(\zeta_t) \le f(\xi) - \epsilon \} \right) \\
& \le \log \left( \sum_{\xi \in \widetilde{\mathcal{X}}} P_t\left( f(\zeta_t) \le f(\xi) - \epsilon \right) \right) \\
& \le \log \left| \widetilde{\mathcal{X}} \right| + \sup_{\xi \in \mathcal{X}} \log P_t\left( f(\zeta_t) \le f(\xi) - \epsilon \right).
\end{aligned} \tag{21}$$

Next, we consider the second assertion. By definition of $B_{\delta'}$ and $\theta_t \sim \mathcal{N}(\mu_t, \Sigma_t^{-1})$, we have

$$P_t(\|\theta_t\| \le B_{t,\delta'}) \le 1 - \delta'.$$

Thus, we have

$$\begin{aligned}
P_t(\zeta_t \notin \mathcal{X}^*(\epsilon)) & \le \delta' + P_t(\zeta_t \notin \mathcal{X}^*(\epsilon), \|\theta_t\| \le B_{t,\delta'}) \\
& \le \delta' + P_t(\zeta_t \notin \mathcal{X}^*(\epsilon + h B_{t,\delta'}; \widetilde{\mathcal{X}}_h)).
\end{aligned}$$

Here, the second inequality follows from the Lipchitz condition of the (posterior) reward function $x \mapsto \theta_t \cdot x$ and the definition of $\widetilde{\mathcal{X}}_h$. Noting that by Eq. (21), we have $p_t(\epsilon'; \widetilde{\mathcal{X}}) \le |\widetilde{\mathcal{X}}| \sup_{\xi \in \mathcal{X}} P_t\left( f(\zeta_t) \le f(\xi) - \epsilon' \right)$, we have $P_t(\zeta_t \notin \mathcal{X}^*(\epsilon)) \le u_t(\delta', h; \widetilde{\mathcal{X}}_h)$ (i.e., the first assertion of Lemma D.1).

Next, we prove the second assertion of Lemma D.1 and inequality (3). For each $t \ge 1$, we let $h_t > 0$ and $\widetilde{\mathcal{X}}_{h_t} \subset \mathcal{X}$ be a finite set with $|\widetilde{\mathcal{X}}_{h_t}| = \widetilde{O}(\text{poly}(t))$ (we choose $h_t$ and $\widetilde{\mathcal{X}}_{h_t}$ later in the proof). For simplicity, we denote $\widetilde{\mathcal{X}}_{h_t}$ by $\widetilde{\mathcal{X}}_t$. We also let $\delta_t' \in (0, 1)$ so that $\delta_t' \le \sup_{\xi \in \mathcal{X}} P_t\left( f(\zeta_t) \le f(\xi) - \epsilon \right)$ holds for sufficiently large $t$ and $-\log(\delta_t') = O(\text{poly}(t))$. Since by Eq. (20) and our assumption of the covariance matrices, we have

$$-\sup_{\xi \in \mathcal{X}} \log(P_t\left( f(\zeta_t) \le f(\xi) - \epsilon \right)) = \Theta(t)$$

for example $\delta_t' = \exp(-t^2)$ satisfies these condition. By the tail probability of the chi square distribution (c.f., (Laurent & Massart, 2000, Lemma 1)) and the definition of $B_{t,\delta'}$, we have $B_{t,\delta_t'} = O(\text{poly}(t))$. Therefore, if we take sufficiently large $m \ge 1$, we have

$$\lim_{t \to \infty} h_t B_{t,\delta_t'} = 0,$$

with $h_t = t^{-m}$. By the first assertion of Lemma D.1, we have the following:

$$\limsup_{t\to\infty} \frac{1}{t} \log P_t(\zeta_t \notin \mathcal{X}^*(\epsilon))$$

$$\leq \limsup_{t\to\infty} \frac{1}{t} \log \left( u_t(\delta_t', h_t; \mathcal{X}_{h_t}) \right)$$

$$\leq \limsup_{t\to\infty} \frac{1}{t} \log \left( |\mathcal{X}_t| \sup_{\xi\in\mathcal{X}} P_t \left( f(\zeta_t) \leq f(\xi) - \epsilon \right) + \delta_t' \right)$$

$$\leq \limsup_{t\to\infty} \frac{1}{t} \log |\mathcal{X}_t| + \limsup_{t\to\infty} \frac{1}{t} \sup_{\xi\in\mathcal{X}} \log P_t \left( f(\zeta_t) \leq f(\xi) - \epsilon \right)$$

$$= \limsup_{t\to\infty} \frac{1}{t} \sup_{\xi\in\mathcal{X}} \log P_t \left( f(\zeta_t) \leq f(\xi) - \epsilon \right).$$

Here, the first and second inequalities follow from the first assertion of Lemma D.1, the third inequality follows by the condition of $\delta_t'$ (i.e., $\delta_t' \leq \sup_{\xi\in\mathcal{X}} P_t \left( f(\zeta_t) \leq f(\xi) - \epsilon \right)$), the last equality follows from $\log |\mathcal{X}_t| = O(\text{poly}(\log t))$. Our assertion Eq.(3) follows from this inequality and Eq. (20). We also have the second assertion of Lemma D.1. This completes the proof. $\qquad\square$

## F.3. Convexity of the Objective Function

The convexity of the function $\Gamma(\cdot; \xi; \zeta, f)$ follows from the lemma below. Since the function $\Gamma^*(\cdot; \zeta, f)$ is defined as the supremum of functions $(\Gamma(\cdot; \xi; \zeta, f))_\xi$, we see that $\Gamma^*(\cdot; \zeta, f)$ is also convex.

**Lemma F.2.** *For any $a \in \mathbb{R}^d$, the function $V \mapsto a^\top V^{-1} a$ defined on $\mathcal{S}_{\succ 0}^d$ is convex.*

*Proof.* We note that the following holds.

$$\sup_{\lambda\in\mathbb{R}^d} a^\top \lambda - 2^{-1}\lambda^\top V\lambda = 2^{-1}a^\top V^{-1}a.$$

As a function of $V$, the LHS is a supremum of affine functions. Therefore, it is a convex function with respect to $V$. $\qquad\square$

## F.4. Some Lemmas for the Convergence Analysis

### F.4.1. SUBGRADIENT

Since the function $\Gamma^*(V; \zeta, \mu)$ is the (pointwise) supremum of a collection of differentiable functions, we can compute its subgradient as follows (cf. (Hiriart-Urruty & Lemaréchal, 2004, Chapter D, Section 4.4)).

**Lemma F.3.** *Let $\zeta \in \mathcal{X}$, $\mu : \mathcal{X} \to \mathbb{R}$, $V \in \mathcal{S}_{>0}^d$ be a symmetric positive semi-definite matrix, and $\xi = \operatorname{argmax}_{\xi\in\mathcal{X}} \Gamma(V, \zeta, \xi; \mu)$. Then, the following is a subgradient of $\Gamma^*(V; \zeta, \mu)$ at $V$: $s_V := -(\epsilon + \mu(\zeta) - \mu(\xi))^{-2} V^{-1}(\zeta - \xi)(\zeta - \xi)^\top V^{-1}$, that is, $\Gamma^*(U; \zeta, \mu) \geq \Gamma^*(V; \zeta, \mu) + \operatorname{Tr} s_V(U - V), \quad \forall U \in \mathcal{S}_{>0}^d$.*

*Proof.* Let $\xi = \operatorname{argmax}_{\xi\in\mathcal{X}} \Gamma(V, \zeta, \xi; \mu)$. Then, by (Hiriart-Urruty & Lemaréchal, 2004, Chapter D, Section 4.4), the gradient of the differentiable function $\Gamma(V, \zeta, \xi; \mu)$ at $V$ is a subgradient of $\Gamma^*(V; \zeta, \mu)$. Then, the assertion of the lemma follows from the matrix calculus. $\qquad\square$

### F.4.2. APPROXIMATE PROJECTION

As we stated in Section 5.1, Algorithm 2 takes an input $V \in \mathbb{R}^{d\times d}$ and returns a distribution $\widetilde{\pi}$ on $\mathcal{X}$ so that $V(\widetilde{\pi})$ is an approximate projection of $V$ to $\mathcal{V}(\mathcal{X})$ with respect to the Frobenius norm. More formally, we have the following lemma and we can prove it by the analysis of the Frank-Wolfe algorithm (Jaggi, 2013) applied to the function $\mathcal{V}(\mathcal{X}) \ni X \mapsto \|X - V\|_F^2$.

**Lemma F.4.** *Let $W \in \mathbb{R}^{d\times d}$ be a symmetric matrix. For $n \geq 2$, we denote by $\Pi_n(W)$ by $V(\widetilde{\pi}^{(n)})$, where $\widetilde{\pi}^{(n)}$ is an output of Algorithm 2 with input $W$. Then, there exists a universal constant $c > 0$ such that the following statement holds. For any $X \in \mathcal{V}(\mathcal{X})$, we have*

$$\|\Pi_n(W) - X\|_F^2 + \|\Pi_n(W) - W\|_F^2 \leq \|X - W\|_F^2 + \frac{c\bar{k}^2}{n+2}.$$

*Or equivalently,*

$$(X - \Pi_n(W)) \cdot (\Pi_n(W) - W) \geq -\frac{c\overline{k}^2}{2(n+2)}.$$

*Proof.* We define a function on $\mathcal{V}(\mathcal{X})$ by $g(X) = \|X - V\|_{\mathrm{F}}^2$ on $\mathcal{V}(\mathcal{X})$ for $X \in \mathcal{V}(\mathcal{X})$. Then, the gradient of $g$ is given as $\langle \nabla g(X), X' \rangle = 2 \operatorname{Tr}(X - V)X'$. Since $\mathcal{V}(\mathcal{X})$ is the convex closure of $\{xx^\top : x \in \mathcal{X}\}$, we note that $\operatorname{argmin}_{X' \in \mathcal{V}(\mathcal{X})} \langle \nabla g(X), X' \rangle$ is equivalent to

$$\operatorname{argmin}_{x \in \mathcal{X}} \operatorname{Tr}(X - V)xx^\top = \operatorname{argmin}_{x \in \mathcal{X}} \|x\|_{X-V}^2.$$

Therefore, Algorithm 2 is the Frank-Wolfe applied for minimizing the function $g$ over $\mathcal{X}$. We also note that the curvature constant $C_f$ (cf. (Jaggi, 2013)) is given as

$$C_g = 2 \sup_{X, X' \in \mathcal{V}(\mathcal{X})} \|X - X'\|_{\mathrm{F}}^2 \leq 2(2\overline{k})^2.$$

Therefore, by (Jaggi, 2013, Theorem 2), there exists a universal constant $c > 0$ such that the following inequality holds:

$$2 \operatorname{Tr}(V(\widetilde{\pi}^{(n)}) - X)(V(\widetilde{\pi}^{(n)}) - V) \leq \frac{c\overline{k}^2}{n+2}, \quad \forall X \in \mathcal{V}(\mathcal{X}).$$

The assertion of the lemma follows from this inequality. $\qquad\square$

### F.4.3. ESTIMATION ERROR

In Algorithm 1, we compute estimations of the values of $f$ and $\zeta$. Therefore, we introduce a concentration inequality for $\mu_t(x)$. Since $f$ is a linear function on $\mathcal{X}$ with probability 1, the following proposition follows from the standard result (Abbasi-Yadkori et al., 2011). However, due to the non-standard setting (i.e., $f$ is a random function), we apply the standard concentration inequality (Abbasi-Yadkori et al., 2011) after sampling a random function $f$ (more formally, we apply the standard concentration inequality on the probability space $(\Omega_{\mathrm{n}}, \mathcal{F}_{\mathrm{n}}, P_{\mathrm{n}})$ for each $\omega_{\mathrm{gp}} \in \Omega_{\mathrm{gp}}$ where notation is defined in Appendix E). We also remark that a similar concentration inequality is proved in the Gaussian process setting (Srinivas et al., 2010).

**Lemma F.5.** *For any $\delta > 0$, the following inequality holds with probability at least $1 - \delta$: $|f(x) - \mu_t(x)| \leq \beta_t(\delta)\sigma_t(x), \quad \forall x \in \mathcal{X}, t \geq 1$, where $\sigma_t(x) = \sqrt{x^\top V_t^{-1} x}$ and $\beta_t(\delta) = \sqrt{\lambda} \left( \sqrt{d \log\left(\frac{1+t\sqrt{k}}{\delta}\right)} + \|\theta^f\| \right)$.*

The following lemma states that $V(\overline{\pi}_t)$ can be approximated by the empirical covariance matrix and can be proved by the standard concentration inequality (Abbasi-Yadkori et al., 2011).

**Lemma F.6.** *Assume $x_t \in \mathcal{X}$ is sampled from a probability kernel $\pi_t$. Then, for any $\delta \in (0,1)$, the following inequality holds with probability at least $1 - \delta$ for any $t \geq 1$, $\left\| \frac{1}{t} \sum_{s=1}^t x_s x_s^\top - V(\overline{\pi}_t) \right\|_{\mathrm{F}} \leq \beta_t'(\delta)/\sqrt{t}$, where $\overline{\pi}_t = \frac{1}{t} \sum_{s=1}^t \pi_s$ and $\beta_t'(\delta) = d\sqrt{(1+t^{-1})\left(1 + 2\log(d^2(1+t)^{1/2}/\delta)\right)}$.*

*Proof.* For each $x \in \mathcal{X}$ and $1 \leq i, j \leq d$, the $(i,j)$ entry of $xx^\top$ is bounded as

$$(xx^\top)_{ij} = \operatorname{Tr} e_i^\top xx^\top e_j = \operatorname{Tr} xx^\top e_j e_i^\top \leq \|xx^\top\|_{\mathrm{F}} \|e_j e_i^\top\|_{\mathrm{F}} \leq \overline{k}.$$

Thus, for each $(i,j)$, $(x_s x_s)_{ij} - V(\pi_s)_{ij}$ is conditionally $2\overline{k}$-subgaussian. By (Abbasi-Yadkori et al., 2011, (11)) and taking a union bound for $(i,j) \in [d] \times [d]$, we have

$$|\sum_{s=1}^t x_s x_s^\top - \sum_{s=1}^t V(\pi_s)|_\infty \leq \sqrt{(1+t)\left(1 + 2\log(d^2(1+t)^{1/2}/\delta)\right)}.$$

$\qquad\square$

We can deduce the following lemma by Lemma F.5.

**Lemma F.7.** *Let $(x_t)_t$ be a sequence of actions selected by Algorithm 1 and $\zeta_t$, $\mu_t$ be as in Algorithm 1. Then, there exists $t_0(\delta) = t_0(\delta; d, \alpha, \pi_{\exp})$ such that, with probability at least $1 - \delta$, we have*

$$|f(x) - \mu_t(x)| \leq \beta_t''(\delta) t^{\alpha/2 - 1/2}, \tag{22}$$

*if $t \geq t_0$, where $\beta_t''(\delta) = \beta_t(\delta/2) \sqrt{2(1-\alpha) \lambda_{\min}^{-1}(V(\pi_{\exp}))}$.*

*Proof.* By Lemma F.6 and the Hoffman-Wielandt theorem (Hoffman & Wielandt, 2003), with probability at least $1 - \delta$, we have

$$\left| \lambda_{\min} \left( \frac{1}{t} \sum_{s=1}^{t} x_s x_s^\top \right) - \lambda_{\min}(V(\bar{\pi}_t)) \right| \leq \beta_t'(\delta) \sqrt{t},$$

We put $\alpha_t := \frac{1}{t} \sum_{s=1}^{t} s^{-\alpha}$. Since $V(\bar{\pi}_t) \succeq \alpha_t V(\pi_{\exp})$, we have

$$\alpha_t \left( \lambda_{\min}(V(\pi_{\exp})) - \beta_t'(\delta) \alpha_t^{-1} t^{-1/2} \right) \leq \lambda_{\min} \left( \frac{1}{t} \sum_{s=1}^{t} x_s x_s^\top \right).$$

We take $t_0 = t_0(\delta; d, \alpha, \pi_{\exp})$ so that $\beta_t'(\delta/2) \alpha_t^{-1} t^{-1/2} < \lambda_{\min}(V(\pi_{\exp}))/2$ if $t \geq t_0$. Then, if $t \geq t_0$, we have

$$\frac{1}{2} \lambda_{\min}(V(\pi_{\exp})) \alpha_t \leq \lambda_{\min} \left( \frac{1}{t} \sum_{s=1}^{t} x_s x_s^\top \right).$$

Thus, for any $x \in \mathcal{X}$, if $t \geq t_0$, the following inequality holds with probability at least $1 - \delta/2$:

$$\sigma_t(x) \leq \sqrt{2} \lambda_{\min}^{-1/2}(V(\pi_{\exp})) \alpha_t^{-1/2} t^{-1/2} \leq \sqrt{2} \sqrt{1-\alpha} \lambda_{\min}^{-1/2}(V(\pi_{\exp})) t^{\alpha/2 - 1/2}.$$

Then, the assertion of the lemma follows from Lemma F.5. $\qquad \square$

Finally, by above results (Lemmas F.5 to F.7), we can upper bound the estimation error of the objective function as follows.

**Lemma F.8.** *Let $(x_t)_t$ be a sequence of actions selected by Algorithm 1 and $\zeta_t$, $\mu_t$ be as in Algorithm 1. Assume $V \succeq t^{-\alpha} V(\pi_{\exp})$. Then, there exists $t_1(\delta) = t_1(\delta; \epsilon, d, \alpha, \pi_{\exp}) \geq t_0(\delta)$ such that with probability at least $1 - \delta$, we have*

$$|\Gamma^*(V; \zeta_t, \mu_t) - \Gamma^*(V; \zeta_\infty, f)|$$
$$\lesssim \epsilon''^{-2} \lambda_{\min}^{-1}(V(\pi_{\exp})) t^\alpha \sqrt{k} \|\zeta_t - \zeta_\infty\| + 2\epsilon''^{-3} \lambda_{\min}^{-1}(V(\pi_{\exp})) \overline{k} t^\alpha \left( B \|\zeta_t - \zeta_\infty\| + \beta_t''(\delta) t^{\frac{\alpha-1}{2}} \right).$$

*Here, $\epsilon'' := f(\zeta_\infty) - \sup_{x \in \mathcal{X}} f(x) + \epsilon > 0$ and the notation $\lesssim$ hides universal constants.*

*Proof.* Let $\mathcal{E}_\delta$ be an event on which the inequality (22) holds if $t \geq t_0(\delta)$. We assume that $\mathcal{E}_\delta$ hold. Then, for $\xi \in \mathcal{X}$, we have

$$\left| \|\zeta_t - \xi\|_{V^{-1}} - \|\zeta_\infty - \xi\|_{V^{-1}} \right| \leq \|\zeta_t - \zeta_\infty\|_{V^{-1}}$$
$$\leq t^{\alpha/2} \lambda_{\min}^{-1/2}(V(\pi_{\exp})) \|\zeta_t - \zeta_\infty\|. \tag{23}$$

For $\xi, V$, we define a function $\varphi : \mathbb{R}^2 \to \mathbb{R}$ as

$$\varphi(p, q) = \frac{(\|\zeta_\infty - \xi\|_{V^{-1}} + p)^2}{(\varepsilon + f(\zeta_\infty) - f(\xi) + q)^2}.$$

Letting $p_t = \|\zeta_t - \xi\|_{V^{-1}} - \|\zeta_\infty - \xi\|_{V^{-1}}$, $q_t = \mu_t(\zeta_t) - f(\zeta_\infty) - \mu_t(\xi) + f(\xi)$ and the Taylor theorem, we have

$$\Gamma(V, \xi; \zeta_t, \mu_t) = \varphi(p_t, q_t)$$
$$= \Gamma(V, \xi; \zeta_\infty, f) + \frac{\partial \phi}{\partial p}(\widetilde{p}_t, \widetilde{q}_t) p_t + \frac{\partial \phi}{\partial q}(\widetilde{p}_t, \widetilde{q}_t) q_t,$$

where $(\widetilde{p}_t, \widetilde{q}_t) = (lp_t, lq_t)$ with $0 \leq l \leq 1$. We take $t_1(\delta) = t_1(\delta; \epsilon'', \nu, B, d, \alpha, \pi_{\exp}) \geq t_0(\delta)$ so that $|q_t| \leq B\|\zeta_t - \zeta_\infty\| + \beta_t''(\delta)t^{(\alpha-1)/2} < \epsilon''/2$ if $t \geq t_1(\delta)$, where $\epsilon'$ is defined as

Then, if $t \geq t_1(\delta)$, we have $|q_t| \leq \epsilon''/2$, and

$$\left|\frac{\partial \phi}{\partial p}(\widetilde{p}_t, \widetilde{q}_t)\right| \lesssim \epsilon''^{-2}\lambda_{\min}^{-1/2}(V(\pi_{\exp}))t^{\alpha/2}\left(\sqrt{k} + \|\zeta_t - \zeta_\infty\|\right) \lesssim \epsilon''^{-2}\lambda_{\min}^{-1/2}(V(\pi_{\exp}))t^{\alpha/2}\sqrt{k},$$

$$\left|\frac{\partial \phi}{\partial q}(\widetilde{p}_t, \widetilde{q}_t)\right| \lesssim \epsilon''^{-3}\lambda_{\min}^{-1}(V(\pi_{\exp}))t^{\alpha}\left(\sqrt{k} + \|\zeta_t - \zeta_\infty\|\right)^2 \lesssim \epsilon''^{-3}\lambda_{\min}^{-1}(V(\pi_{\exp}))t^{\alpha}\overline{k}$$

Here, notation $\lesssim$ hides universal constants. Thus, we have

$$\begin{aligned}
&|\Gamma^*(V; \zeta_t, \mu_t) - \Gamma^*(V; \zeta_\infty, f)| \\
&\lesssim \epsilon''^{-2}\lambda_{\min}^{-1/2}(V(\pi_{\exp}))t^{\alpha/2}\sqrt{k}|p_t| + \epsilon''^{-3}\lambda_{\min}^{-1}(V(\pi_{\exp}))t^{\alpha}\overline{k}|q_t| \\
&\lesssim \epsilon''^{-2}\lambda_{\min}^{-1}(V(\pi_{\exp}))t^{\alpha}\sqrt{k}\|\zeta_t - \zeta_\infty\| \\
&\quad + 2\epsilon''^{-3}\lambda_{\min}^{-1}(V(\pi_{\exp}))\overline{k}t^{\alpha}\left(B\|\zeta_t - \zeta_\infty\| + \beta_t''(\delta)t^{\frac{\alpha-1}{2}}\right).
\end{aligned}$$

$\square$

## F.5. Proof of Proposition D.3

*Proof of Proposition D.3.* We follow the standard proof of the (projected) subgradient method (Ruszczyński, 2006, Chapter 7). However, in our setting, we only have an estimation of the objective function, and we can compute the projection only approximately. Thus, we provide a proof for completeness. In the proof, we define

$$\begin{aligned}
\overline{\Gamma}_t^*(V) &:= \Gamma^*((1 - t^{-\alpha})V + t^{-\alpha}V(\pi_{\exp}); \zeta_t, \mu_t), \\
\Gamma_t^*(V) &:= \Gamma^*(V; \zeta_t, \mu_t).
\end{aligned}$$

for any $V \in \mathcal{V}(\mathcal{X})$. Then, by definition, we have $\Gamma_t^*(V_t) = \overline{\Gamma}_t^*(\widetilde{V}_t)$. For symmetric matrices $A, B \in \mathbb{R}^{d \times d}$, we denote $\operatorname{Tr} AB$ by $A \cdot B$. We assume the event $\mathcal{E}_\delta$ on which Lemma F.8 holds. Let $V^* \in \operatorname{argmin}_{V \in \mathcal{P}(\mathcal{X})} \Gamma^*(V; \zeta, f)$. We let

$$d_t := g_t \cdot (\widetilde{V}_t - V^*)/\|g_t\|_{\mathrm{F}}, \quad U_t := V^* + d_t g_t/\|g_t\|_{\mathrm{F}}.$$

$$\begin{aligned}
\|\widetilde{V}_{t+1} - V^*\|_{\mathrm{F}}^2 &= \|\Pi_n(W_{t+1}) - V^*\|_{\mathrm{F}}^2 \\
&\leq \|W_{t+1} - V^*\|_{\mathrm{F}}^2 + \frac{c\overline{k}^2}{2(n_t + 2)} \\
&= \|\widetilde{V}_t - \eta_t c_t g_t - V^*\|_{\mathrm{F}}^2 + \frac{c\overline{k}^2}{2(n_t + 2)} \\
&= \|\widetilde{V}_t - V^*\|_{\mathrm{F}}^2 - 2c_t\eta_t g_t \cdot (\widetilde{V}_t - V^*) + \eta_t^2 c_t^2 \|g_t\|_{\mathrm{F}}^2 + \frac{c\overline{k}^2}{2(n_t + 2)} \quad (24) \\
&\leq \|\widetilde{V}_t - V^*\|_{\mathrm{F}}^2 - 2c_t\eta_t\|g_t\|(d_t - \eta_t/2) + \frac{c\overline{k}^2}{2(n_t + 2)}. \quad (25)
\end{aligned}$$

Here, the first inequality follows from Lemma F.4 and the second inequality follows from $c_t\|g_t\|_{\mathrm{F}} \leq 1$. Since $g_t$ is a subgradient of $\overline{\Gamma}_t^*$ at $\widetilde{V}_t$ by Lemma F.3, and by definition of $d_t, U_t$, we have

$$\overline{\Gamma}_t^*(U_t) \geq \overline{\Gamma}_t^*(\widetilde{V}_t) + g_t \cdot (U_t - \widetilde{V}_t) = \overline{\Gamma}_t^*(\widetilde{V}_t) = \Gamma_t^*(V_t). \quad (26)$$

If $\liminf_{t\to\infty} d_t = 0$, then there exists a subsequence $\mathcal{T} = \{t_n\}_{n\geq 1}$ such that $\lim_{n\to\infty} t_n = \infty$ and $\lim_{n\to\infty} U_{t_n} = V^*$. we have for $t \in \mathcal{T}$,

$$
\begin{aligned}
\Gamma^*(V^*) &= (\Gamma^*(V^*) - \Gamma^*(U_t)) + \left(\Gamma^*(U_t) - \overline{\Gamma}_t^*(U_t)\right) + \overline{\Gamma}_t^*(U_t) \\
&\geq (\Gamma^*(V^*) - \Gamma^*(U_t)) + \left(\Gamma^*(U_t) - \overline{\Gamma}_t^*(U_t)\right) + \Gamma_t^*(V_t) \\
&= (\Gamma^*(V^*) - \Gamma^*(U_t)) + \left(\Gamma^*(U_t) - \overline{\Gamma}_t^*(U_t)\right) + (\Gamma_t^*(V_t) - \Gamma^*(V_t)) + \Gamma^*(V_t). \quad (27)
\end{aligned}
$$

By $\alpha < \max(1/3, \nu)$, Lemma F.8, and the definition of $\overline{\Gamma}_t^*$, the second and third terms in (27) converge to zero since Lemma F.8 holds uniformly for $V \in \mathcal{V}(\mathcal{X})$. For $t \in \mathcal{T}$, the first term converges zero. Thus, by inequality (27), we have $\Gamma^*(V^*) \geq \liminf_{t\in\mathcal{T}} \Gamma^*(V_t)$. Thus, we have our assertion if $\liminf_{t\to\infty} d_t = 0$. Next, we assume $\liminf_{t\to\infty} d_t > 0$. Since $\lim_{t\to\infty} \eta_t = 0$, we see that there exists $a > 0$ and $t_1 \geq 1$ such that $d_t - \eta_2/2 > a$ for any $t \geq t_1$. By summing both sides of (25) for $t_1 \leq t \leq s$, where $s$ is any integer with $s \geq t_1$, we see that

$$
0 \leq \|V_s - V^*\|_{\mathrm{F}}^2 \leq \|V_{t_1} - V^*\|_{\mathrm{F}}^2 - 2a \sum_{t=t_1}^{s} c_t \|g_t\| \eta_t + \sum_{t=t_1}^{s} \frac{c\overline{k}^2}{2(n_t + 2)}.
$$

If $\liminf_{t\to\infty} c_t \|g_t\|_{\mathrm{F}} > 0$, then we can derive a contradiction by the above inequality by taking $s \to \infty$ since $\sum_{t=1}^{\infty} \eta_t = \infty$ and $\sum_{t=1}^{\infty} 1/(n_t + 2) < \infty$. Thus, we assume that $\liminf_{t\to\infty} c_t \|g_t\|_{\mathrm{F}} = 0$. Since $g_t$ is a subgradient of $\overline{\Gamma}_t^*$ at $\widetilde{V}_t$, we have

$$
g_t \cdot (\widetilde{V}_t - V^*) \geq \overline{\Gamma}_t^*(\widetilde{V}_t) - \overline{\Gamma}_t^*(V^*) = \Gamma_t^*(V_t) - \overline{\Gamma}_t^*(V^*).
$$

Since $\|\widetilde{V}_t - V^*\|_{\mathrm{F}}$ is bounded, we have $0 \geq \liminf_{t\to\infty} \Gamma_t^*(V_t) - \overline{\Gamma}_t^*(V^*)$. Since Lemma F.8 holds uniformly for $V \in \mathcal{V}(\mathcal{X})$, we see that $\liminf_{t\to\infty} \Gamma^*(V_t) = \liminf_{t\to\infty} \Gamma_t^*(V_t) \leq \lim_{t\to\infty} \overline{\Gamma}_t^*(V^*) = \Gamma^*(V^*)$.

Therefore, we have proved that for any $\delta \in (0, 1)$ there exists an event $\mathcal{E}_\delta$ with $P(\mathcal{E}_\delta) \leq 1 - \delta$ such that on $\mathcal{E}_\delta$, we have $\liminf_{t\to\infty} \Gamma_t^*(V_t) = \Gamma^*(V^*)$.

We take a sequence $(\delta_n)_{n=1}^{\infty}$ with $\delta_n \in (0, 1)$ and $\lim_{n\to\infty} \delta_n = 0$ and let $\mathcal{E}_n = \mathcal{E}_{\delta_n}$. We define an event $\mathcal{E}$ as $\limsup_{n\to\infty} \mathcal{E}_n = \cap_n \cup_{k\geq n} \mathcal{E}_k$. Then, by Fatou's lemma, we have $P(\mathcal{E}) \geq \limsup_n P(\mathcal{E}_n) \geq \limsup_n (1 - \delta_n) = 1$. Since for any $\omega \in \mathcal{E}$, there exists $n \geq 1$ such that $\omega \in \mathcal{E}_n$ and, we see that $\liminf_{t\to\infty} \Gamma^*(V(\pi_t); \zeta, f) = \tau^*(f; \zeta)$ holds on $\mathcal{E}_n$, it holds on the event $\mathcal{E}$. This completes the proof. $\qquad\square$

### F.6. Proof of Theorem 6.2

For the proof of the theorem, we introduce the following results (Proposition F.9, Lemma F.10). We note that Proposition 6.3 follows from Proposition F.9.

**Proposition F.9.** *Suppose assumptions in Assumption 6.1 hold. We let*

$$
\widetilde{\Gamma}^*(V; \zeta_\infty, f) = \Gamma^*(V; \zeta_\infty, f) + \lambda_V \|V\|_{\mathrm{F}}^2,
$$

*and $V_{\lambda_V}^* \in \mathrm{argmax}_{V\in\mathcal{V}(\mathcal{X})} \widetilde{\Gamma}^*(V; \zeta_\infty, f)$. Then, for any $\delta \in (0, 1)$, we have the following for any $T$ with probability at least $1 - \delta$:*

$$
\widetilde{\Gamma}^* (V(\overline{\pi}_T); \zeta_\infty, f) - \widetilde{\Gamma}^*(V_{\lambda_V}^*; \zeta_\infty, f) \leq \frac{\rho_T}{T}.
$$

*Here, using notation of Lemma F.8, $\rho_T$ is given as*

$$
t_1 (4\epsilon^{-2} \lambda_{\min}^{-1}(\pi_{\exp}) t_1^{-\alpha} \overline{k}^2 + \lambda_V \overline{k}^2) + \left(\frac{c'}{4\alpha + 1} + 2 + c\right) \overline{k}^2 T^{4\alpha + 1/2}
$$

$$
+ \sum_{t=t_1}^{T} \left|\widetilde{\Gamma}_t^*(\widetilde{V}_t; \zeta_\infty, f) - \widetilde{\Gamma}_t^*(\widetilde{V}_t; \zeta_t, \mu_t)\right| + \sum_{t=t_1}^{T} \left|\widetilde{\Gamma}_t^*(V_{\lambda_V}^*; \zeta_\infty, f) - \widetilde{\Gamma}_t^*(V_{\lambda_V}^*; \zeta_t, \mu_t)\right|.
$$

*and $c' = c'(\epsilon, \pi_{\exp}, \lambda_V) := (4\lambda_{\min}^{-1}(V(\pi_{\exp}))\epsilon^{-2} + 2\lambda_V)^2$.*

*Moreover, we have the following a.s.:*

$$
\lim_{t\to\infty} \Gamma^*(V(\overline{\pi}_t); \zeta_\infty, f) = \tau_{\mathcal{X}}^*(f; \zeta_\infty).
$$

*Proof.* Basically, we follow the standard proof of the projected subgradient descent method (Bubeck, 2015). We let $\widetilde{\Gamma}^*(V; \zeta_\infty, f) = \Gamma^*(V; \zeta_\infty, f) + \lambda_V \|V\|_F^2$ and $\widetilde{\Gamma}_t^*(V; \zeta_\infty, f) = \widetilde{\Gamma}^*((1 - t^{-\alpha})V + t^{-\alpha}V(\pi_{\exp}); \zeta_\infty, f)$, $V_{\lambda_V}^* = \operatorname{argmax}_{V \in \mathcal{V}(\mathcal{X})} \widetilde{\Gamma}^*(V; \zeta_\infty, f)$.

We put $g_t = -(1 - t^{-\alpha})(\epsilon + \mu_t(\zeta_t) - \mu_t(\xi_t))^{-2}v_t v_t^\top + 2\lambda_V \widetilde{V}_t \in \mathbb{R}^{d \times d}$. Since $g_t$ is a subgradient of $\widetilde{\Gamma}_t^*(\cdot; \zeta_t, \mu_t)$ at $\widetilde{V}_t$, we have

$$
\begin{aligned}
&\widetilde{\Gamma}_t^*(\widetilde{V}_t; \zeta_t, \mu_t) - \widetilde{\Gamma}_t^*(V_{\lambda_V}^*; \zeta_t, \mu_t) \\
&\leq \operatorname{Tr} g_t(\widetilde{V}_t - V^*) \\
&= \frac{1}{\eta_t} \operatorname{Tr}(\widetilde{V}_t - W_{t+1})(\widetilde{V}_t - V_{\lambda_V}^*) \\
&= \frac{1}{2\eta_t} \left( \|\widetilde{V}_t - W_{t+1}\|_F^2 + \|\widetilde{V}_t - V_{\lambda_V}^*\|_F^2 - \|V_{\lambda_V}^* - W_{t+1}\|_F^2 \right) \\
&= \frac{\eta_t}{2} \|g_t\|_F^2 + \frac{1}{2\eta_t} \left( \|\widetilde{V}_t - V_{\lambda_V}^*\|_F^2 - \|V_{\lambda_V}^* - W_{t+1}\|_F^2 \right).
\end{aligned}
\tag{28}
$$

By Lemma F.4, we have

$$
\|V_{\lambda_V}^* - \widetilde{V}_{t+1}\|_F^2 \leq \|V_{\lambda_V}^* - W_{t+1}\|_F^2 + \frac{c\overline{k}^2}{n_t + 2}.
\tag{29}
$$

Since $V_t' \succeq t^{-\alpha}V(\pi_{\exp})$, we see that

$$
\|v_t v_t^\top\|_F = (\zeta_t - \xi_t)^\top (V_t')^{-2}(\zeta_t - \xi_t) \leq 4\lambda_{\min}^{-1}(V(\pi_{\exp}))\overline{k}t^{2\alpha}.
$$

Thus, we have

$$
\|g_t\|_F^2 \leq t^{4\alpha}\overline{k}^2 c',
$$

where $c' = c'(\epsilon, \pi_{\exp}, \lambda_V) := (4\lambda_{\min}^{-1}(V(\pi_{\exp}))\epsilon^{-2} + 2\lambda_V)^2$. By this inequality and (28), (29), we have

$$
\begin{aligned}
&\sum_{t=1}^{T} \left( \widetilde{\Gamma}_t^*(\widetilde{V}_t; \zeta_t, \mu_t) - \widetilde{\Gamma}_t^*(V_{\lambda_V}^*; \zeta_t, \mu_t) \right) \\
&\leq 2^{-1}c'\overline{k}^2 \sum_{t=1}^{T} t^{4\alpha}\eta_t + 2^{-1}\eta_T^{-1}\|\widetilde{V}_1 - V_{\lambda_V}^*\|_F^2 + \eta_T^{-1}c\overline{k}^2 \sum_{t=1}^{T} \frac{1}{n_t + 2} \\
&\leq 2^{-1}c'\overline{k}^2 \sum_{t=1}^{T} t^{4\alpha}\eta_t + (2 + c)\overline{k}^2\eta_T^{-1}.
\end{aligned}
$$

Here, we take $n_t$ so that $\sum_{t=1}^{\infty} 1/(n_t + 2) \leq 1$ and take $\eta_t = t^{-2\alpha - 1/2}$. Then, we have

$$
\sum_{t=1}^{T} \left( \widetilde{\Gamma}_t^*(\widetilde{V}_t; \zeta_t, \mu_t) - \widetilde{\Gamma}_t^*(V_{\lambda_V}^*; \zeta_t, \mu_t) \right) \leq \left( \frac{c'}{4\alpha + 1} + 2 + c \right) \overline{k}^2 T^{4\alpha + 1/2}.
\tag{30}
$$

Noting that $\widetilde{\Gamma}_t^*(V; \zeta_\infty, f) \leq 4\epsilon^{-2}\lambda_{\min}^{-1}(\pi_{\exp})t^{-\alpha}\overline{k}^2 + \lambda_V \overline{k}^2$ for $V \in \mathcal{S}_{\succ 0}^d$, for $t \leq t_1$, we have

$$
\sum_{t=1}^{T} \left( \widetilde{\Gamma}^*((1-t^{-\alpha})\widetilde{V}_t + t^{-\alpha}V(\pi_{\exp}); \zeta_\infty, f) - \widetilde{\Gamma}^*(V_{\lambda_V}^*; \zeta_\infty, f) \right)
$$

$$
\leq \sum_{t=1}^{T} \left( \widetilde{\Gamma}_t^*(\widetilde{V}_t; \zeta_\infty, f) - \widetilde{\Gamma}_t^*(V_{\lambda_V}^*; \zeta_\infty, f) \right)
$$

$$
\leq t_1(4\epsilon^{-2}\lambda_{\min}^{-1}(\pi_{\exp})t_1^{-\alpha}\overline{k}^2 + \lambda_V\overline{k}^2) + \sum_{t=t_1}^{T} \left( \widetilde{\Gamma}_t^*(\widetilde{V}_t; \zeta_\infty, f) - \widetilde{\Gamma}_t^*(V_{\lambda_V}^*; \zeta_\infty, f) \right).
$$

$$
\leq t_1(4\epsilon^{-2}\lambda_{\min}^{-1}(\pi_{\exp})t_1^{-\alpha}\overline{k}^2 + \lambda_V\overline{k}^2) + \left( \frac{c'}{4\alpha+1} + 2 + c \right)\overline{k}^2 T^{4\alpha+1/2}
$$

$$
+ \sum_{t=t_1}^{T} \left| \widetilde{\Gamma}_t^*(\widetilde{V}_t; \zeta_\infty, f) - \widetilde{\Gamma}_t^*(\widetilde{V}_t; \zeta_t, \mu_t) \right| + \sum_{t=t_1}^{T} \left| \widetilde{\Gamma}_t^*(V_{\lambda_V}^*; \zeta_\infty, f) - \widetilde{\Gamma}_t^*(V_{\lambda_V}^*; \zeta_t, \mu_t) \right|
$$

$$
=: \rho_T.
$$

Here, the third inequality follows from (30). Since $\widetilde{\Gamma}^*(\cdot; \zeta_\infty, f)$ is a convex function, we have

$$
\widetilde{\Gamma}^* \left( (1-\alpha_T)\overline{V}_T + \alpha_T V(\pi_{\exp}); \zeta_\infty, f \right) - \widetilde{\Gamma}^*(V_{\lambda_V}^*; \zeta_\infty, f) \leq \frac{\rho_T}{T},
$$

where $\overline{V}_T = \frac{1}{T}\sum_{t=1}^{T} \widetilde{V}_t$ and $\alpha_T = \frac{1}{T}\sum_{t=1}^{T} t^{-\alpha}$. Thus, we have the first assertion. Letting $\lambda_V = 0$, by Lemma F.8 and the assumptions, with probability at least $1-\delta$, we have $\lim_{t\to\infty} \Gamma^*(V(\overline{\pi}_t); \zeta_\infty, f) = \tau_{\mathcal{X}}^*(f; \zeta_\infty)$. By the same argument as in the proof of Proposition D.3, we see that $\lim_{t\to\infty} \Gamma^*(V(\overline{\pi}_t); \zeta_\infty, f) = \tau_{\mathcal{X}}^*(f; \zeta_\infty)$. holds a.s. Thus, we have the second assertion. This completes the proof. $\qquad \square$

**Lemma F.10.** *Let $\{(x_t, y_t)\}_t$ be a sequence by running Algorithm 1. Then, for any $x \in \mathcal{X}$, we have $\lim_{t\to\infty}\mu_t(x) \to f(x)$ almost surely.*

*Proof.* This follows from Lemma F.7 and the same argument as in the proof of Proposition D.3. $\qquad \square$

*Proof of Theorem 6.2.* The statement of Theorem 6.2 follows from Lemma 4.1, Lemma F.10, and Proposition F.9. $\qquad \square$

### F.7. Proof of Propositions 6.4 and 6.5

*Proof of Proposition 6.5.* Suppose $\mathcal{X}$ is a unit sphere and let $f(x) = \mu \cdot x$. Then $\zeta_\mu^* = \mu/\|\mu\|$ and by

$$
f(\zeta_\mu^*) - f(x) = \|\mu\| \left( 1 - \frac{\mu}{\|\mu\|} \cdot x \right) = \frac{\|\mu\|}{2} \left\| \frac{\mu}{\|\mu\|} - x \right\|^2 = \frac{\|\mu\|}{2} \|\zeta_\mu^* - x\|^2
$$

the assumption (9) is satisfied with $\kappa_{\mathcal{X}} = 2$.

More generally, let us suppose that for any $x_0 \in \mathcal{X}$ there exists a neighborhood $\mathcal{U}_{x_0} \subset \mathbb{R}^d$ of $x_0$ and an analytic function $\phi_{x_0} : \mathcal{U}_{x_0} \to \mathbb{R}$ such that $\mathcal{X} \cap \mathcal{U}_{x_0}$ is given as $\{x \in \mathcal{U}_{x_0} : \phi_{x_0}(x) = 0\}$ and the minimal eigenvalue of the Hessian of $\phi_{x_0}$ at $x_0$ is larger than a positive constant $c > 0$ independent of $x_0$. We define a (possibly infinite) constant $\gamma(\mu)$ by

$$
\gamma(\mu) = \sup_{x \in \mathcal{X} \setminus \{\zeta_\mu^*\}} \frac{\|x - \zeta_\mu^*\|^2}{|\mu \cdot \zeta_\mu^* - \mu \cdot x|}.
$$

If $\gamma(\mu) < \infty$ for $\mu \neq 0$, then we have the condition (9). We let $\gamma(\mu, x) = \frac{\|x-\zeta_\mu^*\|^2}{|\mu \cdot \zeta_\mu^* - \mu \cdot x|}$. By assumptions there exists an open neighborhood $\mathcal{U} \ni \zeta_\mu^*$ and an analytic function $\phi : \mathcal{U} \to \mathbb{R}$ such that locally $\mathcal{X}$ is given as $\phi(x) = 0$. Since $\zeta_\mu^*$ is an optimal arm, there exists a Lagrange multiplier $\lambda_\mu \in \mathbb{R}$ such that $\mu = \lambda_\mu \nabla\phi(\zeta_\mu^*)$. By $\|\mu\| \neq 0$ and $\|\nabla\phi(\zeta_\mu^*)\|$ is bounded, we see that $\lambda_\mu \gtrsim 1$. Thus, it follows that

$$
|\mu \cdot \zeta_\mu^* - \mu \cdot x| = \lambda_\mu |\nabla\phi(\zeta_\mu^*) \cdot (x - \zeta_\mu^*)| \gtrsim |\nabla\phi(\zeta_\mu^*) \cdot (x - \zeta_\mu^*)|,
$$

for any $x \in \mathcal{X} \cap \mathcal{U}$. By the power series expansion of $\phi$ around $\zeta_\mu^*$, for each $x \in \mathcal{X} \cap \mathcal{U}$, we have

$$0 = \phi(x) = \nabla\phi(\zeta_\mu^*) \cdot (x - \zeta_\mu^*) + \frac{1}{2}(x - \zeta_\mu^*)^\top H(\zeta_\mu^*)(x - \zeta_\mu^*) + R_3(x - \zeta_\mu^*; \zeta_\mu^*).$$

Here, $H(\zeta_\mu^*)$ is the Hessian of $\phi$ at $\zeta_\mu^*$ and $R_3(\cdot; \zeta_\mu^*)$ is the residual of degree greater than or equal to three of the power series expansion. Thus, for any $x \in \mathcal{X} \cap \mathcal{U}$, we have

$$\gamma(\mu, x) = \lambda_\mu^{-1} \frac{\|x - \zeta_\mu^*\|^2}{|\mu \cdot \zeta_\mu^* - \mu \cdot x|} \lesssim \frac{\|x - \zeta_\mu^*\|^2}{(x - \zeta_\mu^*)^\top H(\zeta_\mu^*)(x - \zeta_\mu^*) + 2R_3(x - \zeta_\mu^*; \zeta_\mu^*)}. \tag{31}$$

By the assumption on the Hessian, we see that $\sup_{x \in \mathcal{X} \cap \mathcal{U}} \gamma(\mu, x) < \infty$. If $x \in \mathcal{X} \setminus \mathcal{U}$, then by the uniqueness of $\zeta_\mu^*$, we have $\inf_{x \in \mathcal{X} \setminus \mathcal{U}} \mu \cdot (\zeta_\mu^* - x) > 0$. Therefore, $\sup_{x \in \mathcal{X} \setminus \mathcal{U}} \gamma(\mu, x) < \infty$. Thus, we have $\gamma(\mu) < \infty$. $\square$

*Proof of Proposition 6.4.* By the assumption $\lambda_{\min}(V_t) \gtrsim t^{-\alpha}$, we see that by the same proof, statements of Lemmas F.6 and F.7 hold for the sampling rule $(x_t)_{t \geq 1}$. Let $\beta_t'(\delta)$ and $\beta_t''(\delta)$ be constants defined in these lemmas. By (22) and the definition of $\zeta_t$, we have

$$f(\zeta) \leq \mu_t(\zeta) + \beta_t''(\delta)t^{\alpha/2-1/2} \leq \mu_t(\zeta_t) + \beta_t''(\delta)t^{\alpha/2-1/2}.$$

Similarly, we have

$$\mu_t(\zeta_t) \leq f(\zeta_t) + \beta_t''(\delta)t^{\alpha/2-1/2} \leq f(\zeta) + \beta_t''(\delta)t^{\alpha/2-1/2}.$$

Thus, we have $|f(\zeta) - \mu_t(\zeta_t)| \leq \beta_t''(\delta)t^{\alpha/2-1/2}$. From the assumption (9), $\|\zeta - \zeta_t\| \leq \beta_t^{(3)}(\delta)t^{(\alpha-1)/(2\kappa_\mathcal{X})}$. Here, $\beta_t^{(3)}(\delta) = (c'_{\mathcal{X},f})^{1/\kappa_\mathcal{X}}(\beta_t''(\delta))^{1/\kappa_\mathcal{X}}$. Thus, the assumption (7) holds with $\nu = (1 - \alpha)/(2\kappa_\mathcal{X})$. $\square$

## G. Appendix to Experiments

### G.1. More Details of the Experiments

We provide more details of the experimental results in Section 7. The experiment has been conducted using AMD EPYC 7352 CPU with 64GB RAM. In the experiments, we use the following parameters of the algorithm: $\alpha = 1$, $\eta_t = c_\eta/\sqrt{t}$, $n_t = 1/t^2$, $\lambda_V = 0$, where $c_\eta$ is defined so that $\text{Tr }\eta_1 g_1 = 1$. We take $\pi_{\exp}$ as the uniform distribution on $\mathcal{X}$. For, step sizes $(\gamma_i)_i$ for Algorithm 2, we did a line-search to minimize the objective $\gamma \mapsto \|(1-\gamma)V(\tilde{\pi}_{i-1}) + \gamma a_i^\top a_i - W\|_{\text{F}}^2$. We take a larger $\alpha$ in Assumption 6.1 since we consider a mixture distribution with $\pi_{\exp}$ to ensure $V(\pi_t)$ is positive definite. If this condition is satisfied during the optimization, it is not necessary to consider a mixture distribution and taking a smaller $\alpha$ makes the convergence slower. Thus, we set $\alpha$ a larger value in this experiment. In Algorithm 2, we break the for loop if $\|V(\tilde{\pi}^{(i)}) - V(\tilde{\pi}^{(i-1)})\|_\infty$ is less than a threshold, and we take the threshold as $10^{-3}$. To optimize the fractional objective, we used the Dinkelbach's algorithm (Dinkelbach, 1967). In the implementation, we stop the iteration loop if $|q_n - q_{n-1}| < 10^{-5}$ or $n > 30$, where $q_n$ is given in Section 5.3. We empirically solved the linear and quadratic objective on $\mathcal{X}$ by using the optimization library provided by the SciPy (Virtanen et al., 2020).

### G.2. Additional Experimental Results

As we detailed in Section 7, we have conducted experiments in the following setting; $\mathcal{X}$ is defined as a subset $\{(\cos(\theta), \sin(\theta)) : \theta \in [\theta_0, \theta_1]\}$ of the unit circle in $\mathbb{R}^2$, and $f(x) = (\cos(\theta_f), \sin(\theta_f)) \cdot x$, $\lambda = 10^{-2}, \epsilon = 10^{-2}$, where $\theta_f = a\pi, \theta_0 = 0, \theta_1 = b\pi$, and $a = 0.0, 0.2, \ldots, 1.8$, $b = 0.0, 0.2, \ldots, 1.8$ with $a < b$. Due to the page limit, we only show a subset of the experimental results. In this section, we introduce additional experimental results. In , We show experimental results with $(a, b) = (0.0, 0.2), (0.0, 0.4), (0.0, 0.6), (0.0, 0.8), (0.0, 1.0), (0.0, 1.2), (0.0, 1.4), (0.0, 1.6), (0.0, 1.8), (0.2, 0.4), (0.2, 0.6), (0.2, 0.8), (0.2, 1.0), (0.2, 1.2), (0.2, 1.4), (0.2, 1.6), (0.2, 1.8), (0.4, 0.6)$ in Figure 3, that with $(a, b) = (0.4, 0.8), (0.4, 1.0), (0.4, 1.2), (0.4, 1.4), (0.4, 1.6), (0.4, 1.8), (0.6, 0.8), (0.6, 1.0), (0.6, 1.2), (0.6, 1.4), (0.6, 1.6), (0.6, 1.8), (0.8, 1.0), (0.8, 1.2), (0.8, 1.4), (0.8, 1.6), (0.8, 1.8), (1.0, 1.2)$ in Figure 4, that with $(a, b) = (1.0, 1.4), (1.0, 1.6), (1.0, 1.8), (1.2, 1.4), (1.2, 1.6), (1.2, 1.8), (1.4, 1.6), (1.4, 1.8), (1.6, 1.8)$ in Figure 5. These figures indicate that our proposed method overall outperforms the baselines. Although these figures shows our method overall outperforms the baselines, we see that in some problem instances (such as $(a, b) = (0.0, 1.6)$), MVR outperforms our method and we suspect that in some problem instances, MVR happens to be nearly optimal. Improving an empirical

*Table 2.* CPU time of each method (including time for evaluation)

| method | Uniform | MVR | Ours |
|---|---|---|---|
| time (sec) | 1.2e+00 (1.9e-02) | 1.3e+00 (2.0e-02) | 2.6e+00 (4.3e-02) |

*Table 3.* CPU time of each method (excluding time for evaluation)

| method | Uniform | MVR | Ours |
|---|---|---|---|
| time (sec) | 8.6e-03 (6.7e-05) | 1.9e-01 (6.4e-04) | 1.5e+00 (2.3e-02) |

performance of our proposed method (for example, by selecting an appropriate step size $\eta_t$) is also a future work of this study.

In Tables 2, 3, we show cpu time for each algorithm for the left most instance in Figure 2. In these tables, the number represents time in seconds for running one experiment and shows the mean (std) over 10 repetitions. Table 2 shows the numbers including cpu time for computing $p_t$ (the evaluation metric) and Table 3 shows the number excluding cpu time for computing $p_t$. These results show our method took about 1.5 seconds while MVR took 0.2 seconds. This is natural since the instance-independent baselines (Uniform and MVR) do not solve the optimization objective over the space of probability measures. The experimental results show our method runs in reasonable time.

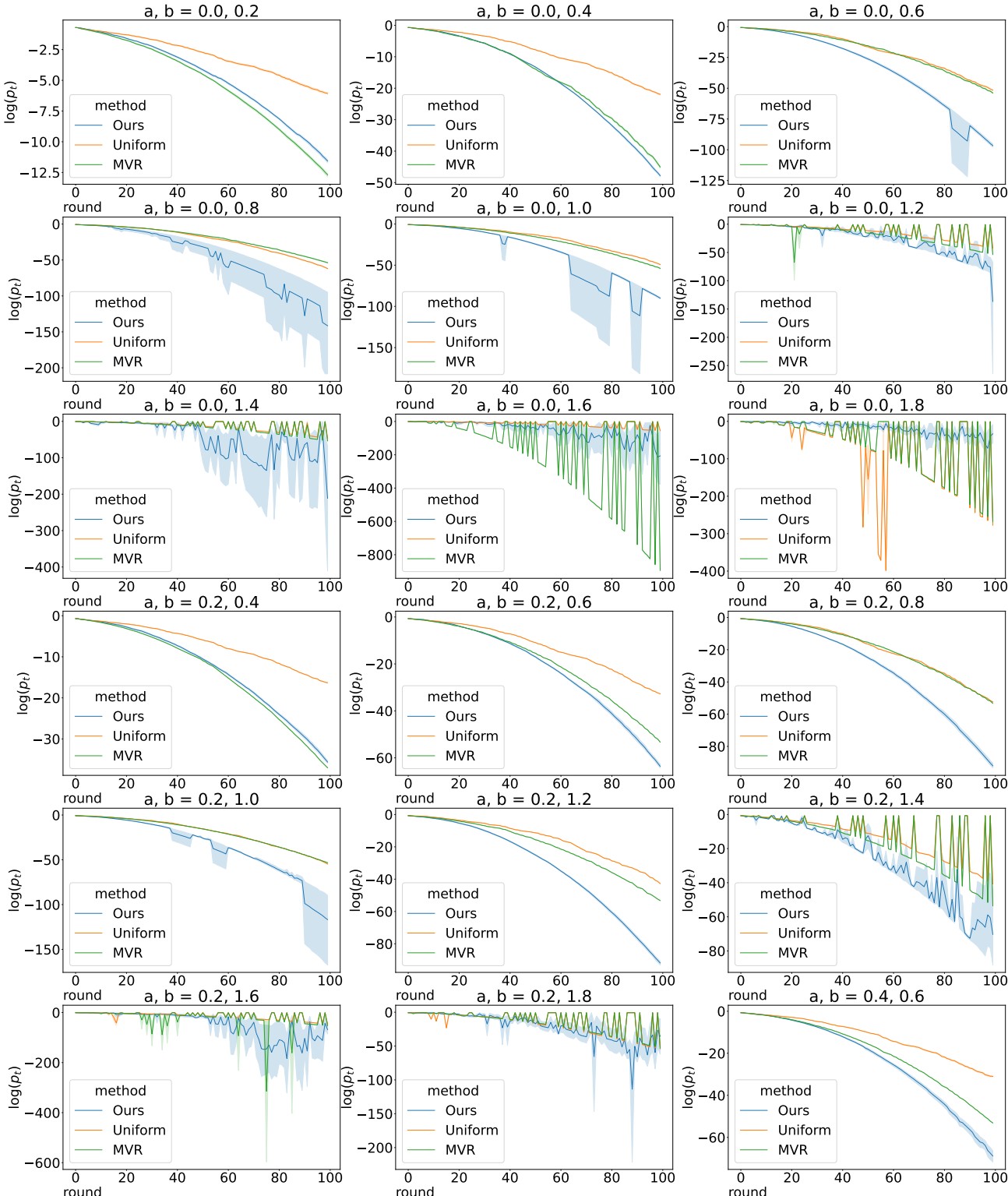

*Figure 3.* Experimental Results for $(a, b) = (0.0, 0.2), (0.0, 0.4), (0.0, 0.6), (0.0, 0.8), (0.0, 1.0), (0.0, 1.2), (0.0, 1.4), (0.0, 1.6), (0.0, 1.8), (0.2, 0.4), (0.2, 0.6), (0.2, 0.8), (0.2, 1.0), (0.2, 1.2), (0.2, 1.4), (0.2, 1.6), (0.2, 1.8), (0.4, 0.6)$.

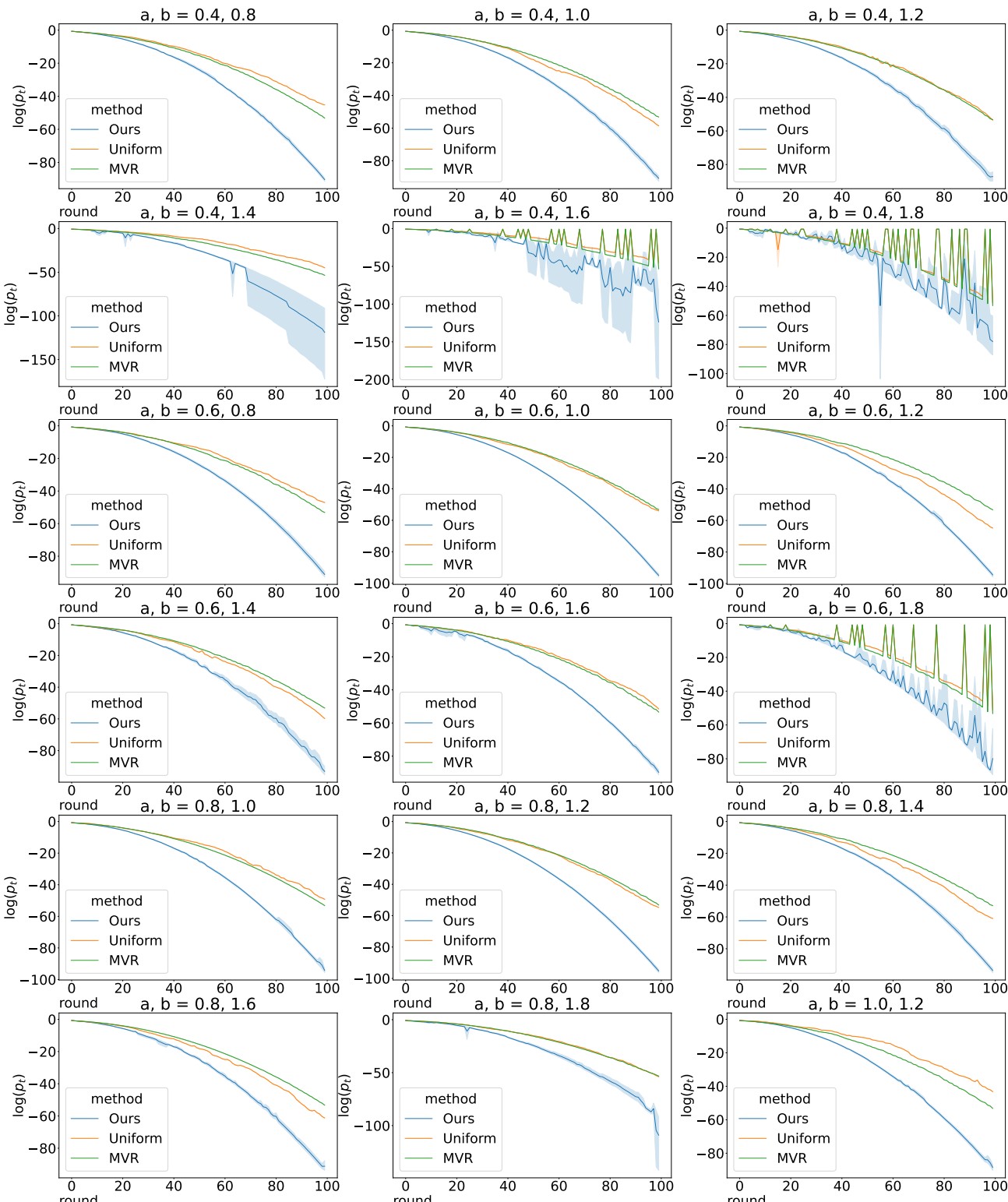

*Figure 4.* Experimental Results for $(a, b) = (0.4, 0.8), (0.4, 1.0), (0.4, 1.2), (0.4, 1.4), (0.4, 1.6), (0.4, 1.8), (0.6, 0.8), (0.6, 1.0), (0.6, 1.2), (0.6, 1.4), (0.6, 1.6), (0.6, 1.8), (0.8, 1.0), (0.8, 1.2), (0.8, 1.4), (0.8, 1.6), (0.8, 1.8), (1.0, 1.2).$

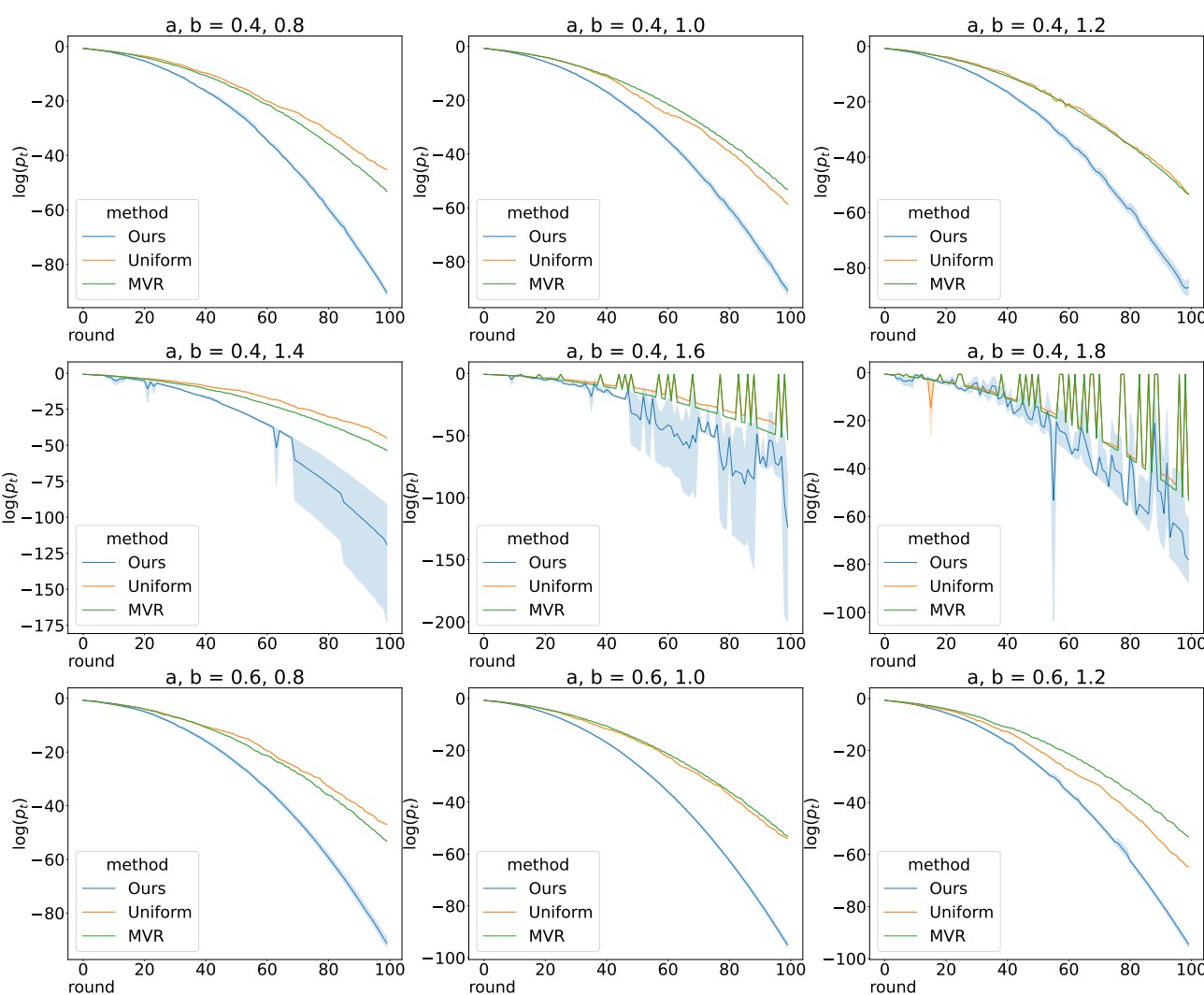

*Figure 5.* Experimental Results for $(a, b) = (1.0, 1.4), (1.0, 1.6), (1.0, 1.8), (1.2, 1.4), (1.2, 1.6), (1.2, 1.8), (1.4, 1.6), (1.4, 1.8), (1.6, 1.8)$.

