# OpenReview forum: "Instance-Optimal Pure Exploration for Linear Bandits on Continuous Arms"
_ICML.cc/2025/Conference — ICML 2025 poster_

### Official Review · Reviewer_oQBX · 2025-02-24

**Overall Recommendation:** 3

**Summary:**

This paper studies $\epsilon$-BAI for Bayesian linear bandits with Gaussian noise and Gaussian prior on compact and continuous arm sets. The metric of performance that is being minimized is the posterior probability of identifying an $\epsilon$-optimal arm conditioned on the history of the observations. In particular, they consider instance-dependent asymptotic rate of convergence of this conditional posterior probability. This setting is challenging due to the infinite dimensionality of the space of policy (i.e., continuous distribution on the set of arms) and the non-smoothness of the objective. To alleviate these challenges, the authors re-parametrize the optimization problem with respect to the design matrix being induced by the sampling policy. As a consequence, they need to extract a policy from the optimized design matrix, which is done with projection and reconstruction following along the lines of the approximate Caratheodory problem. This requires solving quadratic and fractional quadratic optimization problem on the arm set. The authors provide an upper and lower bound on the conditional posterior probability of error. Moreover, they numerically compare their algorithm with other benchmarks on synthetic instances.

**## update after rebuttal**
Including the discussions will improve the paper in its revised version, hence I raised my score towards weak accept.

**Claims And Evidence:**

I. Tractable algorithm.

While PCMA aims at solving more tractable optimization problems, the discussion on the tractability of PCMA is not fully convincing based on Sections 5.2 and 5.3. To the best of my understanding, PCMA has both a large computational and space cost. In particular, it would be valuable to have a more detailed discussion on the

Quadratic Objective. What is the per-call computational and space cost of the quadratic objective used in Algorithm 2 ?  More importantly, the authors mention that “some problems can be solved in a polynomial time, and some problems are NP-hard”. Is the quadratic objective solvable in polynomial time for the problems considered in this paper, or is it NP-hard ? Even if the computational cost is polynomial, this oracle is being called a superlinear number of times at each iteration, i.e., $O(t^{u})$ with $u> 1$. Therefore, it seems to be a major computational bottleneck.

Quadratic Fractional Objective. What is the per-iteration computational and space cost of the quadratic fractional objective (being called at each time step) ? What is the convergence rate, and how does it impact the choice of the “break” condition for this optimization procedure ? In particular, Appendix G.1 seems to suggest that at most 30 iterations are used, and refer to a quantity $q_n$ not defined in Section 5.3.
Algorithm 2. The memory requirement seems also to be superlinear with time. Is it possible to maintain a sparser approximation or the policy ?
If the discussion is not possible for general set of arms, then it would be enlightening to illustrate the discussion in some special cases such as (1) the unit sphere or (2) a spherical cap as considered in Section 7.

Based on the small scale of the experiments and the lack of numerical computational costs, I am wondering whether PCMA can be truly considered as a tractable and practical algorithm.

II. Instance-Dependent Optimality.

While the authors claim instance-dependent optimality, their actual statement doesn’t reflect what is commonly referred to as “Instance-Dependent Optimality”. Therefore, it would be better to be more precise and discuss the difference with what is usually referred to as asymptotic optimality.

Theorem 6.2 is a statement on the conditional posterior probability of misidentification, hence it doesn’t account for the randomness of the observations themselves.

Theorem 6.2 assumes knowledge of the unknown characteristic time $\tau^\star$ in order to match the optimal rate. This is not a practical assumption, hence the theorem doesn’t hold for practical implementation

III. Recommendation Rule.

The authors state their result for a given stream of recommendation rule. However, to the best of my understanding, they have only theoretical guarantees (Section 6.3) and experiments (Section 7) for the greedy recommendation rule. The paper would gain in clarity by being specialized for the greedy recommendation rule. The greedy recommendation rule is by far the least computational expensive method, even though it is not asymptotically optimal on all instances.

Section 4.3 glimpses at a more sophisticated notion of recommendation rule, i.e., the (instantaneous) furthest answer in Jourdan & Degenne (2022). However, this choice is far from being tractable. Even with the approximated algorithms from PCMA, it would require an additional outer optimization loop for a non-convex optimization problem. On top of the intractability of this procedure, it will probably be challenging to show that Equation (7) holds for this choice.

IV. Stopping Rule.

To the best of my understanding, based on paragraph “Stopping Rule” in Section 3 and Section 4.2, the authors do not provide a clear and convincing discussion as regards the stopping rule. In particular, the authors should explicitly define what is the stopping rule that they are referring to, compare it to the literature on stopping rule, and explain whether this stopping rule has some theoretical guarantees.

In particular, while the authors only control the posterior probability conditioned on $\mathcal F_t$, proving $(\epsilon,\delta)$-PAC also require controlling the randomness of the observations themselves, i.e., without the conditioning on $\mathcal F_t$. Therefore, comparing their upper bound to $\delta$ will not be enough to obtain an $(\epsilon,\delta)$-PAC stopping rule, both theoretically and empirically.

The upper bound proposed in Lemma 4.2 does not seem to be truly non-asymptotic. It is based on a discretization of the space (Lemma D.1) and the different terms do not seem to be explicit.

**Essential References Not Discussed:**

To the best of my knowledge, most relevant literature is discussed. Nevertheless, more details could be added in order to compare the obtained results for continuous set of arms with the ones known for discrete set of arms.

**Experimental Designs Or Analyses:**

The empirical metric of performance used in the experiments appears to be unusual, and lack details on its practical implementation. Instead of considering the empirical proportion of error as a function of time, the authors consider the conditional posterior probability of misidentification. Since it is hard to compute, they consider as proxy the upper bound from Lemma 4.2, yet dropping the terms $a_t$ and $b_t$. Due to the optimization over the continuous set of arms, it is not clear how $p_t$ is actually computed numerically. Are the authors using a discretization of the space, e.g., the one from Lemma D.1 ? Even for a discrete set, it is not clear how each term is numerically computed. Is it based on the Gaussian cumulative distribution discussed at the end of Section 4.2.


While promising, the experimental results provided in Section 7 and Appendix G seem to be preliminary. A more extensive empirical evaluation would be appreciated. In particular, the empirical claims would be strengthened by having:
- More than one hundred rounds for each run, and more than ten runs for each instance.
- Plots of the empirical proportion of error as a function of time.
- Plots of the actual computational cost (in terms of CPU time) for the proposed algorithms to highlight that it is tractable.
- Bayesian experiments in order to actually match the Bayesian setting considered in this paper.
- Number of samples needed for the approximated upper bound (from Lemma 4.2) on the conditional posterior probability of error to be lower than $\delta$.
- Ablation studies to better understand the impact of the hyper-parameter of the algorithm. In particular, the experiments use $\lambda_{V} = 0$, yet the theory require $\lambda_{V}$ to depend on the unknown characteristic time (Theorem 6.2 and Propostion 6.3).

**Methods And Evaluation Criteria:**

See “Experimental Designs Or Analyses” section for details on the empirical evaluation.

**Other Comments Or Suggestions:**

Equation (16) in Appendix F.2 lacks a square on $r_t$ in the first term of the right hand side.

**Other Strengths And Weaknesses:**

I. Lack of clarity.

In general, the paper would gain in clarity by considering a specific recommendation rule instead of studying a general formula under some assumptions. This obfuscates the discussion on the main challenges of the continuous set of arms. It would be valuable to clearly state what are the theoretical challenges in proving Theorem 6.2. Is a direct extension of the analysis for discrete sets ?  Moreover, the paper would benefit from more examples to illustrate the bounds and the algorithms, e.g. special case of the unit sphere. Currently, it is difficult to parse everything in all its generality.

II. Confusing notation for the conditional posterior probability of misidentification.

To the best of my understanding, the only randomness in the conditional posterior probability of misidentification is with respect to $f_t \sim P_t$ since $\zeta_t$ is given. The dependency in $f_t$ is not made explicit and seems hidden in the notation $\mathcal X^*(\epsilon)$ which should depend on $f_t$ and not on $f$ fixed at initialization.

**Questions For Authors:**

Several questions have been asked in the previous sections.

**Relation To Broader Scientific Literature:**

The authors wrote a wrong statement on the literature:
“Jedra & Proutiere (2020) [...] showed that an optimal sampling policy is the round-robin manner using the orthonormal basis.”
Jedra & Proutiere (2020) show that round-robin sampling yields an algorithm whose sample complexity is order-wise optimal. Order-wise means that it has the same scaling as the optimal characteristic time, however the lower and upper bound are not matching asymptotically. Therefore, round-robin is not an optimal sampling policy, only one that is “not too bad”.

**Theoretical Claims:**

Lemma 4.1 seems unusual for the literature due to its dependency in $t$ on both sides of the equations. In particular, while both the sampling and the recommendation rules are not explicitly given and satisfy some general conditions, the upper and lower bounds are matching. This seems too good to be true. Could the authors provided more discussion ?

Lower bound from (2). While it could be independent of the sampling policy as in Theorem 2.1 in Li et al (2024) and Theorem 1 in Russo (2016), the analysis should most likely require more assumptions as regards the recommendation rule at each time. For example, a minimum requirement should be that the recommendation is $\epsilon$-optimal for $\mu_t$, which is satisfied by the greedy recommendation rule. It seems like this property is used in the equation 15 in Appendix F.2. Some statements should also be detailed more in the proof as they don’t seem to be trivial: (Line 756-757) “convergence of t in LHS of (18) is uniform [...] we can exchange lim and sup”.

Upper bound from (3). This seems to be very surprising that it holds for general  sampling policy without assumptions on its asymptotic behavior. Intuitively, one should at least require convergence of the sampling policy. While this assumption is not made explicit in Lemma 4.1, it seems used in Appendix F.2: (Line 715) “we assume $(\pi_t)$ converges,”.

---

> ### Author Rebuttal · Authors · 2025-03-31
>
> We sincerely appreciate the reviewer's thorough and insightful review and apologize for any statements that may have caused confusion.
>
> ## Tractability of the algorithm
> First, we note that our method is tractable in terms of the number of oracle calls. We do not specify algorithms (optimization oracles) for the quadratic and fractional quadratic objectives.
>
> ### Computation Oracles
> Please refer to the response to Reviewer 4oes (we note that $q\_{n}$ is defined in Line 317 (right)).
>
> ### Space complexity
> In every iteration of Alg. 1, we approximate our sampling policy using Alg. 2. The choice of $n_t = t^u$ is motivated by the convergence rate of the Frank-Wolfe method (Lemma F.4). Under certain assumptions, the Frank-Wolfe method exhibits a faster convergence rate [Garber & Hazan, 2015], and the cardinality of the support of $\pi_t$ can be significantly reduced. However, this typically requires the set $\mathcal{V}(\mathcal{X})$ to be strongly convex, and we are unaware of any arm-sets that satisfy this assumption.
>
> ## Instance-Dependent Optimality
> We agree that our method is asymptotically optimal. However, as we discussed in the introduction, our analysis is instance-dependent unlike existing methods in the continuous arm setting (such as MVR).
> Theorem 6.2 assumes knowledge of the unknown characteristic time, however, as we remarked after Prop 6.2, we provided a convergence analysis in the case when $\lambda\_{V} = 0$ in Prop D.2.
>
> ## Recommendation Rule
> We agree that our main result, Thm. 6.2, relies on assumption (7) regarding the recommendation rule, and we provide sufficient conditions for (7) only for the greedy recommendation rule. To improve clarity, in the introduction of the revised version of this paper, we will discuss this.
>
> ## PAC with respect to the unconditional probability
> As clarified in the paper, we consider the $\epsilon$-BAI problem under the Bayesian reward model setting. Proving that the method is $(\epsilon, \delta)$-PAC with respect to the unconditional probability $P$ is not a primary focus of this study. We believe there are several ways to theoretically validate a pure exploration algorithm. For instance, under a Bayesian setting, Russo 2016 also provided a similar analysis (asymptotic bound of the posterior error probability) to this study, but it is considered to be an important literature in this field.
>
> Moreover, the core ideas of PMCA can be applied to existing algorithms (such as LεBAI (Jourdan & Degenne, 2022)), since they do not clarify the optimization method regarding the sampling distribution, and it would be possible to prove the extended algorithm is PAC.
>
> ## Stopping Rule
> For any explicitly computable upper bound $u\_t$ of the conditional probability of misidentification and $\delta \in (0, 1)$, we refer to the stopping rule by the stopping time defined as the minimum $t$ satisfying $u\_{t} \le \delta$. Assuming $\mathcal{X} \subset [0, 1]^d$, we make the statement of Lemma D.1 more explicit. Using notation of 2 of Lemma D.1, by the tail probability of chi squared distribution, we can take $h\_{t} = t^{-(1 + c)}$ and $|\mathcal{X}\_{h}|$ is given as $(\sqrt{d}h^{-1})^d$. Then, $u\_t(\delta'\_t, h\_t, \mathcal{X}\_{h\_t})$ gives an explicit upper bound. As we discussed in Line 240, $\sup\_{\xi \in \mathcal{X}} P\_t(\cdots)$ can be solved using the fractional quadratic objective.
>
>
> ## Lemma 4.1
> Both sides of Lemma 4.1 are defined asymptotically, therefore they do not depend on $t$. In addition, unlike (Russo 2016), both sides depend on asymptotic behavior of general recommendation rule $\zeta\_t$ and (the mean of) general sampling rule $\pi\_{t}$ rather than the optimal one (our analysis can also be regarded as a generalization of Eq. (1) of Glynn & Juneja (2004)). In Eq. (15) in Appendix, we use the standard fact on the normal distribution that unconditionally holds for the signature of $\epsilon + \mu\_t(\zeta\_t) - \mu\_t(\xi\_t)$. Moreover, in the proof (around Line 715), we assume a **sub-sequence** of $(V(\overline{\pi}\_{t}))\_{t\ge 1}$ converges, and we *do not assume* $V(\overline{\pi}\_{t})$ converges in the statement of Lemma 4.1. We hope this resolves your concern and if you have further questions, we would be happy to answer them.
>
>
> ## Experiments
> We appreciate your suggestions. We refer to the response to Reviewer huXi for CPU time of each method. We have conducted an ablation study of the parameter $\lambda\_{V}$ for the second ($a, b=0.2, 0.6$) problem instance in Fig 2, where MVR is sub-optimal. The table shows the mean (standard deviation) of our evaluation metric. We found that while the original choice ($\lambda\_{V} = 0$) outperforms the baselines, the optimal choice of $\lambda\_{V}$ is around $1e^{-1}$.
>
> | $\lambda\_{V}$ | 0.0 | 1e-2 | 1e-1|
> | -- | -- | -- | -- |
> | $log (p\_t)$ | -63.6 (1.4)|-64.1 (3.7)|-69.4 (2.3) |
>
> **References**
> - Garber & Hazan, Faster rates for the Frank-Wolfe method over strongly-convex sets, ICML 2015

---

> > ### Comment · Reviewer_oQBX · 2025-04-02
> >
> > I thank the authors for their thorough and detailed answers, as well as the additional experiments. At the time being, I am inclined to increase my score to weak accept.
> >
> > **Instance-Dependent Optimality**. Given that Theorem 6.2 is obtained by using Proposition 6.3, could the authors discuss the challenges encountered when using Proposition D.2 to generalize Thm 6.2 when $\lambda_{V} = 0$ ? Do the authors believe that a finer analysis could show the convergence in Proposition D.2 for all sequence, i.e., replacing liminf by $\inf$, or is it truly a limitation when taking $\lambda_{V} = 0$ ?
> >
> > **Miscellaneous**. Could the authors explicitly state how $p_t$ is computed in their experiments ?

---

> > > ### Author Response · Authors · 2025-04-03
> > >
> > > We sincerely appreciate the reviewer's further feedback.
> > >
> > >
> > > ## Analysis for Proving Theorem 6.2 Under the Condition $\lambda\_{V} = 0$
> > > Thank you for bringing this problem up. Based on your suggestion, we reconsidered this problem, and we believe a simple modification of our analysis can prove Theorem 6.2 even if $\lambda\_{V} = 0$. If we can confirm this rigorously, we will update our manuscript in the revision. Specifically, the revised statement of Proposition 6.3 would be as follows:
> > >
> > > **Proposition 6.3 (revised version)** Let $(\pi\_t)_{t}$ be the sampling rule of PMCA (with $\lambda\_{V} = 0$), then we have $\lim\_{t \rightarrow \infty}\Gamma^{\ast}(V(\overline{\pi}\_{t}); \zeta\_{\infty}, f) = \tau^{\ast}(f; \zeta\_{\infty})$.
> > >
> > > We introduced the regularizer $\lambda\_{V}$ to ensure the sequence $(V(\pi\_{t}))\_{t}$ converges. However, as we discussed, this condition is not necessary. In the proof of Theorem 6.2, the condition $\lambda\_{V} > 0$ is not necessary until Line 1159. We can prove the above proposition by the inequality at Line 1157. Then, Lemma 4.1 implies the main result (Theorem 6.2).
> > >
> > >
> > > ## Proposition D.2
> > > > Do the authors believe that a finer analysis could show the convergence in Proposition D.2 for all sequence.
> > >
> > > No, we believe that the condition for the step sizes is too general to derive such a result.
> > > We provide more details in the following and we explain the difficulty in deriving Theorem 6.2 from Proposition D.2. In Proposition D.2, we consider a general condition for the step sizes (i.e. $\lim\_{t\rightarrow \infty}\eta\_t = 0$ and $\sum\_{t=1}^{\infty}\eta\_t = \infty$ as in [Ruszczynski, A. P., 2006, Theorem 7.2]). This leads to a weak statement on the convergence result, that is, we have $\liminf\_{t\rightarrow \infty} \Gamma^{\ast}(V(\pi\_t); \zeta\_\infty, f) = \tau^{\ast}(f; \zeta\_\infty)$. We refer to this equality as Eq. (a). To prove our main result (Theorem 6.2), at least we need the following condition $\liminf\_{t\rightarrow \infty} \Gamma^{\ast}(V(\overline{\pi}\_t); \zeta\_\infty, f) = \tau^{\ast}(f; \zeta\_\infty)$. We refer to this equality as Eq. (b). Equations (a) and (b) are slightly different because of $V(\pi\_t)$ and $V(\overline{\pi}\_t)$, where we recall that $\overline{\pi}\_{t} = \frac{1}{t}\sum\_{s=1}^{t}\pi\_{s}$. If $(V(\pi\_t))\_t$ is a convergence sequence, Eq. (a) implies Eq. (b), but in general, we suppose the conclusion of Proposition D.2 is too weak to prove Theorem 6.2. The value of Proposition D.2, apart from the condition on $\lambda_{V}$, is that we have a convergence result under a general condition on the step sizes. We will clarify this in the appendix of the revised paper.
> > >
> > > ## Miscellaneous
> > > To compute $p\_t$ in the experiments, we used the formula introduced at Line 234, which follows from Eq (15) and the monotonicity of the cumulative distribution function $\Phi$. Assume that $\epsilon + \mu\_t(\zeta\_t) - \mu\_t(\xi) \ge 0$ holds for any $\xi \in \mathcal{X}$ (the greedy recommendation rule satisfies this condition). Then, by $\inf\_{\xi \in \mathcal{X}} \frac{\epsilon + \mu\_t(\zeta\_t) - \mu\_t(\xi)}{|\zeta\_t - \xi|\_{\Sigma\_{t}^{-1}}} = (\sup\_{\xi \in \mathcal{X}} \frac{|\zeta\_t - \xi|\_{\Sigma\_{t}^{-1}}^2}{(\epsilon + \mu\_t(\zeta\_t) - \mu\_t(\xi))^2})^{-1/2}$, we can compute $p\_t$ by calling the oracle for the quadratic fractional objective. We will clarify this in the revision.
> > >
> > >
> > > Thank you again for your valuable feedback to improve our manuscript.

---

### Official Review · Reviewer_4oes · 2025-03-06

**Overall Recommendation:** 3

**Summary:**

This paper investigates a pure exploration problem with linear bandit feedback on continuous arm sets, aiming to identify an $\varepsilon$-optimal arm with high probability. Previous approaches for continuous arm sets have employed instance-independent methods, due to technical challenges such as the infinite dimensionality of the space of probability measures, and the non-smoothness of the objective function. This paper designs a novel and tractable algorithm that addresses these challenges by using a reparametrization of the sampling distribution and projected subgradient descent. However, this approach introduces new challenges related to the projection and reconstruction of the distribution from the reparametrization. This paper addresses these by focusing on the connection to the approximate Caratheodory problem. Compared to the original optimization problem on the infinite-dimensional space, the proposed method is tractable, requiring only the solution of quadratic and fractional quadratic problems on the arm set. This paper also provides empirical results.

**Claims And Evidence:**

The claims made in this paper are supported by clear and convincing evidence.

**Essential References Not Discussed:**

None

**Experimental Designs Or Analyses:**

The experiments look reasonable.

**Methods And Evaluation Criteria:**

The proposed methods and evaluation criteria make sense.

**Other Comments Or Suggestions:**

Please see the weaknesses above.

**Other Strengths And Weaknesses:**

Strengths:

1. This paper proposes a tractable algorithm PMCA for Posterior error Minimization for a general Continuous Arm set based on the PSGD (projected subgradient descent method).
2. This paper provides upper and lower bounds to show that the proposed algorithm is asymptotically optimal.
3. The theoretical contribution of this paper is solid and interesting to the linear bandit community.
4. Empirical results are provided to demonstrate the effectiveness of the proposed algorithm.

Weaknesses:

1. The readability of this paper should be improved. This paper is dense and hard to follow.
2. This paper provides asymptotically optimal results. It seems that the practical significance of asymptotically optimal optimality is not strong, because the number of times we sample an option is finite and we care more about the optimality of the final output option in real-world applications. Can the asymptotic results in this paper provide any insight for the finite-time results/optimality?
3. More discussion on the challenges of generalization from the finite-arm set to the continuous arm set is needed.
4. Can you give more concrete examples for the computational oracles used in the proposed algorithm? For example, in what subproblems such computational oracles exist, and what are the corresponding concrete computational oracles (algorithms)?

**Questions For Authors:**

Please see the weaknesses above.

**Relation To Broader Scientific Literature:**

This paper is relevant to the literature.

**Theoretical Claims:**

The theoretical results look reasonable, but I didn’t go through every proof.

---

> ### Author Rebuttal · Authors · 2025-03-31
>
> We sincerely appreciate the reviewer's thorough and insightful review. We will revise our manuscript based on your suggestions.
> ## Readability
> We appreciate reviewer's suggestion. In a revision, we will improve the readability of the paper (e.g. by making detailed statements in Prop 6.5 concise and refer to the Appendix for a more precise statement).
>
> ## Asymptotic optimality
> Lemma 4.2 provides a non-asymptotic upper bound for the posterior probability of misidentification that asymptotically matches the actual conditional probability. However, a more refined analysis guaranteeing non-asymptotic optimality remains an important avenue for future research. We will discuss this limitation in the Conclusion section of the revised paper.
>
> ## More discussion on the challenges of generalization from the finite-armed setting
> While it would be difficult to prove that generalizing from the finite-armed setting is impossible for any existing algorithm, the non-smooth optimization objective over the space of probability measures is a fundamental problem that any asymptotically optimal algorithm must address. To the best of our knowledge, even in the finite-armed setting, the efficient solution of this problem has not been thoroughly discussed. We can also view our contribution as an efficient algorithm for the finite-armed setting, particularly when dealing with a large number of arms.
>
>
> ## Computational Oracles
> ### Quadratic Objective
> For example if the arm set $\mathcal{X}$ is defined as an interval constraint $l \le q(x) \le u$ with a quadratic function $q$ (this includes the case of the unit sphere), the quadratic objective can be solved in polynomial time by the Lagrange relaxation (Park&Boyd, 2017). The relaxed problem is a semi-definite problem (convex problem). If we use an interior point method (barrier method), to obtain an $\epsilon'$-optimal solution, the outer loop needs $O(\log(1/\epsilon'))$ iterations and the convergence rate of the inner loop is linear convergence. Per iteration computational complexity is $O(d^4)$ with $O(d^2)$ space complexity (Boyd&Vandenberghe, 2004, Chapter 11.8.3).
>
> In general, the problem can be NP-hard. We note that MVR (Vakili et al., 2021) has the same problem and similar to the MVR implementation (PosteriorStandardDeviation) in the BoTorch library, we use a non-linear solver (such as L-BFGS-B) with a randomly selected initial point in our experiment.
>
>
> ### Quadratic Fractional Objective
> Let $A(x)/B(x)$ be the objective we want to maximize over $x \in \mathcal{X}$. As we briefly discussed in Sec. 5.2, if we use the Dinkelbach's algorithm, we call the quadratic objective oracle once in each iteration of the Dinkelbach's algorithm. More precisely, if we define $q^{(n + 1)} = A(x^{(n)})/B(x^{(n)})$ and $x^{(n)} = argmax\_{x} A(x) - q\_{n}B(x)$, then $q\_{n}$ converges to the optimal value $q\_{\ast}$ and its convergence rate is superlinear (the error goes to zero faster than any geometric sequence) (Schaible, 1976). Then one can obtain a solution of the original problem by $ argmax\_{x} A(x) - q^{\ast}B(x)$. Due to the superlinear convergence rate, we use a limited number of iterations (i.e., 30) in our experiments.
>
>
>
> **References**
>
> - Boyd & Vandenberghe, Convex Optimization, Cambridge University Press, 2004

---

### Official Review · Reviewer_bZeb · 2025-03-12

**Overall Recommendation:** 4

**Summary:**

This paper investigates the problem of best arm identification (BAI) for Bayesian linear bandits, where the action set is assumed to be continuous. While existing investigations, e.g., [Jedra et. al.] establish optimal algorithms when the set of arms is finite, BAI for linear bandits under infinite actions is hitherto unexplored. This paper takes a step towards solving this problem, specifically, making the following key contributions:

- It introduces an instance-dependent measure (which I will call problem complexity in the rest of the review), a counterpart of its finite-armed setting, and shows that the asymptotic posterior probability of error for any algorithm decays exponentially at a rate proportional to the problem complexity, and no larger (converse result).
- It devises an algorithm which achieves this asymptotic error rate

There are some novel observations, which I would also like to highlight.

- The paper introduces a reparameterization to solve the infinite-dimensional problem (in the probability space) to an equivalent problem in the matrix space, (and hence finite dimensional)
- It introduces a projected gradient descent-based algorithm for BAI (which has been used in prior works), and the projection step is tackled using a novel connection with the approximate Caratheodary theorem.

Overall, the paper takes a step towards solving BAI in the linear bandit setting with continuous arm sets.

**Claims And Evidence:**

Yes, I think that the claims and evidences are coherent and sufficient.

**Essential References Not Discussed:**

N/A

**Experimental Designs Or Analyses:**

This is a theoretical paper; having said that, the experimental settings suffice to bolster theoretical claims.

**Methods And Evaluation Criteria:**

Yes, the experiment settings make sense.

**Other Comments Or Suggestions:**

Answered above.

**Other Strengths And Weaknesses:**

I have already listed the strengths of this paper. A weakness of the paper is that the writing can be improved. Here are instances of typographical / grammatical inconsistencies:

- **Line 95:** It should be $f(\zeta)$
- **Lines 161-162:** that estimates $\zeta_t$ an $\epsilon$-optimal arm ...
-  **Lines 220-222:** the term $b_t$ is given as (an upper bound) the ...
- **Lines 272-274:** similarly should be Similarly

**Questions For Authors:**

I have the following questions for the authors:

- the authors mention that "..the posterior probability $P_t(\zeta_t\in\mathcal{X}^*(\epsilon))$ of misidentification is known to the learner...". How does the learner know $\mathcal{X}^*(\epsilon)$?
- What is the computational complexity of the reparameterization $\pi_t \mapsto \mathbf{V}_t$, since it involves an expectation operator?
- It seems that the expression of the sub-gradient in (6) is its gradient. How do the authors deal with the non-differentiability at points due to the inner supremum, say, in (4)?

**Relation To Broader Scientific Literature:**

- BAI is an important problem in the bandit literature, with various practical applications including A/B testing, clinical trials, recommender systems, etc. Investigating the continuous-armed setting enhances our theoretical understanding of BAI.
- Prior works establish optimal (fixed-confidence and fixed-budget) algorithms for linear bandits. Examples include [Jedra et. al.], [Vakili et. al.]. Most of these investigations consider a finite set of arms.
- This paper positions itself as the one which extends BAI to continuous arm sets, which has not been sufficiently investigated.

**Theoretical Claims:**

I did not have time to check the correctness of the proofs. I would be happy to take a look at any specific part, if any issue is raised.

---

> ### Author Rebuttal · Authors · 2025-03-31
>
> We sincerely appreciate the reviewer's thorough and insightful review. We will revise our manuscript based on your suggestions.
>
> ## Posterior probability of misidentification
> As we briefly discussed in Line 133 (right), the posterior probability $P\_t(\zeta\_t \not \in \mathcal{X}^{\ast}(\epsilon))$ is known to the learner, i.e., it is a $\mathcal{F}\_{t}$-measurable variable because it is defined using the conditional probability $P\_t$. More concretely, we can rewrite $P\_t(\zeta\_t \not \in \mathcal{X}^{\ast}(\epsilon)) = P\_t(\zeta\_t \le \sup_{x}f(x) - \epsilon)$. We note that conditioned on $\mathcal{F}\_t$, the reward function $f$ is identified with $f\_{t}$. In addition, the distribution of $f\_{t}$ is known due to the Bayesian setting and we provide it explicitly in Line 152 (left). Therefore, we can essentially compute the posterior probability of misidentification. We hope this resolves your concern. We will clarify the statements in Line 133 in the revision of this paper.
>
> ## Computational Complexity
> Since in each round, we add $n\_{t}$ points to the support of the sampling distribution, the computational complexity of the reparametrization is given as $O(d^2 n_t)$, which is polynomial in $t$. We appreciate your suggestion and will add the discussion on this in Section 5.3.
>
> ## Subgradient
> $g\_t$ defined in Eq. (6) is a subgradient of the function defined in Eq. (4). We used a known fact that if the function $F$ is defined as the supremum of convex functions $(F\_{i})\_{i \in I} $, then a subgradient of $F$ is given as a gradient of $F\_{i^{\ast}}$, where $i^\ast$ attains the supremum (Hiriart-Urruty & Lemarechal 2004, Chapter D, Lemma 4.4.1). We appreciate the reviewer's suggestion and will briefly explain this in the main paper.

---

### Official Review · Reviewer_huXi · 2025-03-13

**Overall Recommendation:** 3

**Summary:**

This work studies the problem of pure exploration for linear bandits particularly with continuous arms. The paper begins by establishing a lower bound on the asymptotic posterior probability of misidentification, and then proposes a tractable algorithm, called PMCA, for minimizing the error probability. The authors provide theoretical analysis showing that the suggested algorithm finds the optimal sampling rule that achieves the asymptotic optimality for any given recommendation rule. The numerical experiments show that the algorithm outperforms MVR and the uniform sampling strategy, confirming the theoretical finding.

**Claims And Evidence:**

I agree that the suggested algorithm PMCA significantly reduces the computational burden required to optimize the sampling distribution.

However, the term “instance-dependent optimality” sounds considerably misleading. Algorithm 1 and Theorem 6.2 treat the recommendation rule $(\zeta_t)_t$ as something given exogenously. Theorem 6.2 states that the sampling rule of PMCA achieves the asymptotic optimality for any given recommendation rule satisfying some regularity condition, without arguing the optimality of the recommendation rule.

I believe that, conventionally, a pure exploration algorithm corresponds to a combination of sampling rule $\pi$ and recommendation rule $\xi$. It does not make sense to me that an algorithm can be said to be instance-optimal without having the recommendation rule specified.

I guess that the authors are aware of that it is technically challenging to guarantee that both of sampling rule and recommendation rule safely converge to their optimal combination, say $(\pi^*, \zeta^*)$. In this work, some assumptions are being made for one component in order to guarantee convergence of the other component (e.g., Assumption 6.1 for Theorem 6.2 vs. $\lambda_{min}(V_t) \geq t^{-\alpha}$ for Proposition 6.4). It was not discussed whether the assumptions can be satisfied simultaneously. Jedra & Proutiere (2020) had introduced the notion of forced exploration in order to break this mutual dependence. I believe the authors should explicitly discuss this gap.

**Essential References Not Discussed:**

None.

**Experimental Designs Or Analyses:**

- I believe the authors should report the computational time of the algorithms, because it is one of the main contributions of this paper.
- I hope to see performance/speed of the algorithm with discretization.
- In order to highlight the practical importance of PMCA, I would like to recommend the authors to find/test an application with many and densely distributed arms. Possibly, LLMs can help in direction (e.g., an arm corresponds a text's embedding).

**Methods And Evaluation Criteria:**

The proposed algorithm PMCA looks completely sensible.

**Other Comments Or Suggestions:**

- In Proposition 6.3, the term $\zeta^*$ was used without any definition.
- Please see Experimental Designs or Analyses for my suggestions on experiments.

**Other Strengths And Weaknesses:**

Although I am skeptical about the claim “near optimal”, I do appreciate the ideas behind PMCA -- the use of PSGD method and the reparameterization trick. As suggested in Experimental Designs or Analyses, I believe that this algorithm has a great potential for the situations with many many arms.

**Questions For Authors:**

None.

**Relation To Broader Scientific Literature:**

I am not sure how the key contributions can extend beyond the linear bandit literature.

**Theoretical Claims:**

I carefully read the statements provided in the main body, but not the proofs in the appendix. I gently believe that the proofs are rigorous.

---

> ### Author Rebuttal · Authors · 2025-03-31
>
> We sincerely appreciate the reviewer's thorough and insightful review. We will revise our manuscript based your on suggestions.
>
>
> ## Instance-dependent optimality and recommendation rule
> We agree that while an asymptotically optimal algorithm needs both optimal sampling and recommendation rules, we mainly focus on the optimization problem regarding the sampling rules, which we believe is the most challenging aspect in the continuous arm setting. More precisely, our main result, Theorem 6.2, relies on assumption (7) on the recommendation rule, and we show the greedy recommendation rule satisfies (7) under some assumptions. To improve clarity, in the introduction (or conclusion section) of the revised version of this paper, we will explicitly explain this more in detail.
>
> ## Assumption 6.1 and the assumption for Proposition 6.4
> Our method satisfies the condition on $\lambda_{min}(V_t)$ for Proposition 6.4 because we consider a mixture distribution of $(1-t^{-\alpha})\tilde{\pi}\_{t} + t^{-\alpha}\pi\_{exp}$ at Line 15 in Algorithm 1. Thus, we can focus on Assumption 6.1 and we believe that we already discussed the validity of the assumption in Sections 6.1 and 6.3. We will clarify this in the revision and appreciate the reviewer's suggestion.
>
>
> ## Experiments
> We sincerely appreciate reviewer's suggestion on the experiments. In the following table, we show cpu time for each algorithm for the left most instance in Figure 2. Regarding a method that discretizes the arm set, the efficiency of such a method can be arbitrary worse due to the discretization, however, we need different problem instances (higher dimensional problem instances) to demonstrate the inefficiency.
>
> In the following table, the number represents time in seconds for running one experiment and shows the mean (std) over 10 repetitions. (w/ eval_time) represents the number includes cpu time for computing $p_t$ (the evaluation metric) and (w/o eval_time) represents the number excluding cpu time for $p_t$.
>
> The table shows our method took about 1.5 seconds while MVR took 0.2 seconds. This is natural since the instance-independent baselines (Uniform and MVR) do not solve the optimization objective over the space of probability measures. The experimental results show our method runs in a reasonable time.
>
>
> | Uniform (w/ eval_time) | MVR (w/ eval_time) | Ours (w/ eval_time) | Uniform (w/o eval_time) | MVR (w/o eval_time) | Ours (w/o eval_time) |
> | -- | -- | -- | -- | -- | -- |
> |1.2e+00 (1.9e-02) | 1.3e+00 (2.0e-02) | 2.6e+00 (4.3e-02) | 8.6e-03 (6.7e-05) | 1.9e-01 (6.4e-04) | 1.5e+00 (2.3e-02)|
>
>
> ## Practical applications
> As you mentioned, recent papers consider an application of bandits to LLMs [Li, Y., 2025, Nguyen, Quang H., et al., 2024, Jinnai & Ariu, 2024]. For instance, Jinnai & Ariu applied the Correlated Sequential-Halving to the text generation tasks (such as machine translation, text summarization, and image captioning tasks), where the objective is to find a text sequence with the best utility efficiently. By regarding an embedded sequence as an arm, we can formulate it as a BAI problem with a linear reward model. Since the set of possible sequences is exponentially large, our algorithm has potential application to this problem.
>
> More generally, we can extend any linear bandit algorithm to Bayesian optimization algorithms by random Fourier features (or quadrature Fourier features) [Mutný, M., & Krause, A. 2019]. The continuous arm setting naturally arises in the Bayesian optimization setting and has numerous applications including material design and hyper-parameter optimization. As we discussed in the conclusion section, it is an important future study of this paper to directly extend our result to the Bayesian optimization setting (without such a reduction).
>
> **References**
> - Jinnai, Y., & Ariu, K. (2024). Hyperparameter-Free Approach for Faster Minimum Bayes Risk Decoding. In ACL (Findings).
> - Li, Y. (2025). LLM Bandit: Cost-Efficient LLM Generation via Preference-Conditioned Dynamic Routing. arXiv preprint arXiv:2502.02743.
> - Mutný, M., & Krause, A. (2019). Efficient high dimensional bayesian optimization with additivity and quadrature fourier features. Advances in Neural Information Processing Systems 31, 9005-9016.
> - Nguyen, Quang H., et al. (2024). "MetaLLM: A High-performant and Cost-efficient Dynamic Framework for Wrapping LLMs." arXiv preprint arXiv:2407.10834 (2024).
>
>
> ## Response to other comments
> > In Proposition 6.3, the term $\zeta^{\ast}$ was used without any definition.
>
> We thank the reviewer for pointing out this. In Proposition 6.3, $\zeta^{\ast}$ is a typo and should be $\zeta_{\infty}$. We will correct this in the revision.

---

### Decision · Program_Chairs · 2025-05-01

**Decision:**

Accept (poster)

**Comment:**

The paper investigates the problem of identifying an $\epsilon$-optimal arm in linear bandits with a continuous action space. The analysis is conducted within a Bayesian framework, assuming a Gaussian prior over the unknown parameter. The objective is to minimize the posterior probability of error.

Due to the continuous nature of the action space, instance-specific analysis poses significant challenges. To address this, the authors employ a reparametrization of the sampling distribution combined with projected subgradient descent. They introduce PMCA, an algorithm that, for a given recommendation rule, provably achieves asymptotic optimality.

The contributions are substantial, as acknowledged by all reviewers. However, there are several areas for improvement, as outlined in the reviews and the authors’ rebuttal. In particular:

1. The authors should provide a more detailed discussion of the computational and memory complexities of PMCA, and report these metrics in the numerical experiments.

2. The use of the term instance-optimality may be misleading, as pointed out by Reviewers oQBX and huXi. This notion should be clarified.

3. The presentation could benefit from simplification and improved clarity. Reviewers have suggested, for instance, focusing on the greedy recommendation rule to enhance readability.

4. The computational oracles used by the algorithm should be exemplified and discussed in greater detail.

5. A clear and convincing discussion of the stopping rule is needed to complete the methodological picture.